# Large Language Model for Lossless Image Compression with Visual Prompts

## Abstract

Recent advancements in deep learning have driven significant progress in lossless image compression. With the emergence of Large Language Models (LLMs), preliminary attempts have been made to leverage the extensive prior knowledge embedded in these pretrained models to enhance lossless image compression, particularly by improving the entropy model. However, a significant challenge remains in bridging the gap between the textual prior knowledge within LLMs and lossless image compression. To tackle this challenge and unlock the potential of LLMs, this paper introduces a novel paradigm for lossless image compression that incorporates LLMs with visual prompts. Specifically, we first generate a lossy reconstruction of the input image as visual prompts, from which we extract local and global features to serve as visual embeddings for the LLM. The residual between the original image and the lossy reconstruction is then fed into the LLM along with these visual embeddings, enabling the LLM to function as an entropy model to predict the probability distribution of the residual. Extensive experiments on multiple benchmark datasets demonstrate our method achieves state-of-the-art compression performance, surpassing both traditional and learning-based lossless image codecs. Furthermore, our approach can be easily extended to images from other domains, such as medical and screen content images, achieving impressive performance. These results highlight the potential of LLMs for lossless image compression and may inspire further research in related directions.

## 1 Introduction

Lossless image compression aims to reduce image size as much as possible without introducing any distortion, making it essential for high-quality data storage and transmission. Furthermore, the techniques used in lossless compression often play a key role in lossy compression methods. Over the past few decades, numerous effective lossless image codecs have been developed. Among these, traditional codecs such as PNG (Boutell, 1997), WebP (Google, 2010), FLIF (Sneyers & Wuille, 2016), and JPEG-XL (Alakuijala et al., 2019) have achieved strong compression performance through hand-crafted coding algorithms. For example, JPEG-XL employs invertible transforms and a sophisticated context model, including tree structure and pre-context predictor selection, to compress images effectively. In recent years, learning-based lossless image codecs (Mentzer et al., 2019; 2020; Zhang et al., 2021b;a; Bai et al., 2024) become increasingly popular. L3C (Mentzer et al., 2019), for instance, utilizes a hierarchical probability prediction framework and introduces auxiliary latent representations to model the probability distribution of image data. These state-of-the-art (SOTA) methods typically rely on empirical knowledge in image compression and employ meticulously designed models to achieve better compression performance.

Recently, Large Language Models (LLMs) have achieved significant breakthroughs in Natural Language Processing tasks, and their applications have extended to vision tasks, driving substantial progress in areas such as image generation (Ge et al., 2024; Pang et al., 2024) and image restoration (Zheng et al., 2024). The primary objective of LLMs is to predict the probability distribution of the next token in a sequence. Consequently, more advanced LLM results in more precise modeling of data distribution. Similarly, entropy coding in lossless compression seeks to accurately model data distribution to minimize the coding bitrate. This parallel suggests that LLMs could potentially serve as powerful tools for entropy coding.

Recent work by Delétang et al. (2023) supports this perspective, demonstrating that LLMs not only achieve impressive results in text compression but also demonstrate strong potential for lossless image compression. This highlights the advantages of leveraging LLMs in the compression domain. However, pretrained LLMs primarily encapsulate textual prior knowledge, whereas image compression relies more on visual information for optimal performance. Therefore, it is crucial to bridge the gap between the textual nature of LLMs and visual data compression tasks. Unfortunately, the existing approach (Delétang et al., 2023) directly treats the pixel values of input images as indexes for LLMs, overlooking the inherent spatial relationships within the images. Consequently, the compression efficiency of this method is suboptimal. For instance, the model proposed by Delétang et al. (2023) with 7B parameters performs only slightly better than PNG (Boutell, 1997). Thus, how to effectively unlock the prior knowledge of LLMs and activate their potential for lossless image compression remains a critical issue that deserves in-depth exploration.

In this work, we propose a novel framework for lossless image compression that leverages the LLM with visual prompts. Specifically, the image is initially compressed using a lossy codec, and this lossy reconstruction is then employed as visual prompts for the LLM. Subsequently, the LLM is used to predict the probability distribution of the residual between the lossy reconstruction and the original image. Finally, the probability distributions of the residual pixels are modeled using the Gaussian Mixture Model (GMM), where the parameters are predicted from the output features generated by the LLM. Furthermore, by finetuning the pretrained LLM with Low-Rank Adaption (LoRA) (Hu et al., 2021), we further enhance our compression performance. Our approach has been evaluated on several benchmark datasets, including Kodak, CLIC, and DIV2K. The results demonstrate that our method achieves SOTA performance, comparable to other well-designed codecs. Our research provides novel insights into lossless image compression and highlights the potential of LLMs for this task.

Our main contributions can be summarized as follows:

- By employing the lossy reconstruction as visual prompts for the LLM, we guide the LLM for more efficient lossless data compression.
- The extensive experimental results demonstrate the SOTA performance of our approach on benchmark datasets. Moreover, our approach can be readily applied to images from other domains, such as screen content images and medical images.

## 2 RELATED WORK

### 2.1 LOSSY IMAGE COMPRESSION

Lossy image compression methods aim to minimize coding distortion at a given bitrate. Traditional lossy image coding standards, such as JPEG (Wallace, 1991) and BPG (Bellard, 2018), employ manually designed modules to improve the compression performance. For instance, the widely-used JPEG codec leverages the discrete cosine transform (DCT) to reduce spatial redundancy and employs Huffman coding to further reduce bitrates losslessly. Most lossy codecs adhere to the rate-distortion principle, selecting optimal coding modes to achieve better compression performance.

Recent advancements in learning-based lossy image compression (Liu et al., 2023; Jiang & Wang, 2023; Li et al., 2024) have surpassed the SOTA traditional codecs like VVC (Bross et al., 2021). The hyperprior model by Ballé et al. (2018) has been studied as a powerful paradigm, applying lossy transforms, quantization, and efficient lossless encoding of latent representations. Some works (Cheng et al., 2020; Zhu et al., 2022; Zou et al., 2022) employ advanced architectures, such as attention mechanism (Vaswani et al., 2017) and Swin-Transformer (Liu et al., 2021), to improve information retention during lossy transforms. Additionally, studies like Minnen et al. (2018) have optimized the lossless latent coding, incorporating autoregressive components with the hyperprior to capture causal context. Refinements of the context model have led to further improvements in compression (Minnen & Singh, 2020; He et al., 2021; 2022).

Many advancements in hyperprior-based methods focus on enhancing the lossless compression of latent representations by achieving more accurate distribution estimation. Consequently, lossy and lossless image compression are closely related, with lossless compression techniques often contributing to greater efficiency in lossy compression.

## 2.2 Lossless Image Compression

Traditional lossless image codecs, such as PNG (Boutell, 1997), WebP (Google, 2010), FLIF (Sneyers & Wuille, 2016), and JPEG-XL (Alakuijala et al., 2019), typically utilize hand-crafted techniques to reduce intra-image redundancy. These methods typically follow a process of filtering, transforming, quantizing, and applying entropy coding to generate the final bitstream. Recently, learning-based lossless image compression has gained significant attention, typically consisting of two stages: 1) constructing a statistical model to capture the probability distribution of image data. 2) utilizing this statistical model to encode the image into a bitstream using entropy tools such as arithmetic coding (AC) or asymmetric numerical systems (ANS) (Duda, 2013). We employ AC as the lossless data compression technique, due to its widespread use in coding systems and its ability to generate nearly optimal-length codes based on a given probability distribution and input sequence. It encodes an entire message as a single number within the interval [0, 1) (represented in binary), using a probabilistic model to subdivide the interval into subintervals proportional according to each symbol's probability.

To enhance statistical models for lossless image compression, deep generative models have been introduced and can be broadly categorized into three types: 1) *Autoregressive models*, such as PixelRNN (Van Den Oord et al., 2016) and PixelCNN (Van den Oord et al., 2016), which predict pixel distributions based on conditional dependencies with previously obtained pixels via masked convolutions. 2) *Flow models*, such as iVPF (Zhang et al., 2021b) and iFlow (Zhang et al., 2021a), which leverage invertible transforms to simplify latent distributions for efficient entropy coding. 3) *Variational Auto-Encoder (VAE) models*, like L3C (Mentzer et al., 2019), which employ VAE architectures to model image distributions. It is noteworthy that some studies have managed to achieve lossless compression by first compressing the image using a lossy encoder, and then compressing the residuals. For example, RC (Mentzer et al., 2020) integrates BPG for image compression and a CNN for residual compression, whereas DLPR (Bai et al., 2024) combines VAE with autoregressive models to enhance performance.

However, these methods typically rely on complex network designs and are constrained by limited training datasets, especially in the fields like medical images where data is scarce. This highlights the need for a simple pipeline that leverages the extensive prior knowledge embedded in pretrained models from other datasets to enhance compression efficiency.

## 2.3 Large Language Models

Large language models (LLMs) have gained significant attention in natural language processing (NLP) and artificial general intelligence (AGI) for their impressive abilities in language generation, in-context learning, world knowledge, and reasoning (Wang et al., 2023). LLMs can quickly adapt to specific tasks using techniques like Adapters (Houlsby et al., 2019) and Low-Rank Adaptation (LoRA) (Hu et al., 2021). Recent research has extended the potential of LLMs to computer vision tasks, such as image classification and segmentation (Gou et al., 2024; Yang et al., 2023). However, these studies primarily focus on aligning textual and visual semantics while overlooking low-level visual features. Addressing this gap, LM4LV (Zheng et al., 2024) employs LLMs for image restoration, emphasizing their understanding of low-level visual features. Additionally, Delétang et al. (2023) demonstrates that LLMs, when viewed as compressors, can outperform traditional codecs like PNG in lossless image compression, highlighting their potential in this field.

## 3 Methodology

The overall framework of our proposed lossless image compression pipeline is illustrated in Fig. 1. The original image $\mathbf{x}$ is first compressed using a lossy codec, producing a lossy reconstructed image $\mathbf{x}_l$. Then we divide $\mathbf{x}_l$ and the residual image $\mathbf{r}$ into non-overlapping patches of size $p \times p$, denoted as $\{\mathbf{x}_l^1, \ldots, \mathbf{x}_l^N\}$ and $\{\mathbf{r}^1, \ldots, \mathbf{r}^N\}$, where $N$ represents the total number of patches. During the encoding process, each patch is processed independently. We predict the probability distribution of each pixel within a residual patch in an autoregressive manner and encode these pixels using arithmetic coding. For instance, when encoding patch $\mathbf{r}^n$ (where $n = 1, 2, \cdots, N$), the entire lossy reconstruction $\mathbf{x}_l$ and its corresponding lossy reconstructed patch $\mathbf{x}_l^n$ are used as visual prompts to extract visual embeddings for the LLM. The pixels in residual patch $\mathbf{r}^n$ are then autoregressively

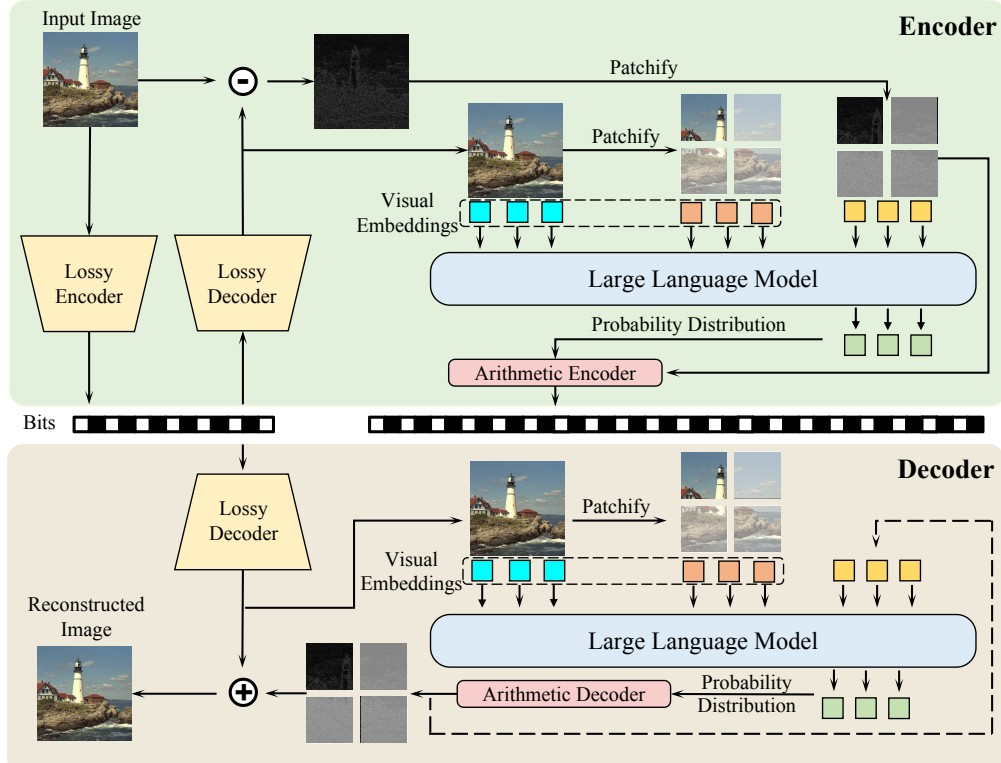

Figure 1: Overview of the encoding and decoding process. A lossy reconstruction $\mathbf{x}_l$ and its patch $\mathbf{x}_l^n$ serve as visual prompts for the LLM to predict the residual's probability distribution, with the decoding process mirroring encoding by generating residual tokens autoregressively.

fed into the LLM to estimate the probability distribution. Given the estimated distributions, we losslessly encode $\mathbf{r}^n$ into a bitstream using arithmetic encoding. The final bitstream comprises the lossy reconstruction $\mathbf{x}_l$ and its corresponding residual image $\mathbf{r}$.

During the decoding procedure, the lossy reconstructed image $\mathbf{x}_l$ is first decoded. Both $\mathbf{x}_l$ and its patch $\mathbf{x}_l^n$ are then utilized as visual prompts to autoregressively obtain the distribution for each pixel in the residual patch $\mathbf{r}^n$. Finally, the full residual image is decoded, and the original image is reconstructed by combining the lossy reconstruction $\mathbf{x}_l$ with the residual image $\mathbf{r}$. It is important to note that for the lossy codecs, we can either choose a traditional compression method or employ an end-to-end learned compression method. Here we use BPG (Bellard, 2018) as the default lossy codec.

## 3.1 INPUT EMBEDDINGS

In existing LLMs, the tokenizer converts text into corresponding indexes, which are then used to obtain embeddings through an embedding layer. For image compression task, Delétang et al. (2023) proposes using pixel values directly as indexes and reusing the embeddings originally trained for text dataset. However, this approach may not fully capture the relationships within the image domain, and the mismatch between textual embeddings and image pixel values may lead to poor performance. Moreover, the prompt technique, which is crucial for large language models, has been overlooked in Delétang et al. (2023).

To address the aforementioned challenges, we introduce visual prompts and visual embeddings as illustrated in Fig. 2. For compressing a residual patch, the visual prompts consist of two components: global lossy image and local lossy patch. To extract global embeddings $\mathbf{z}_g \in \mathbb{R}^{k_g \times d}$, we design a simple Global Embedding Module that utilizes several convolutional layers to capture pixel relationships from $\mathbf{x}_l$. For the local embeddings $\mathbf{z}_l^n \in \mathbb{R}^{p^2 \times d}$ of patch $n$, we directly use pixel values

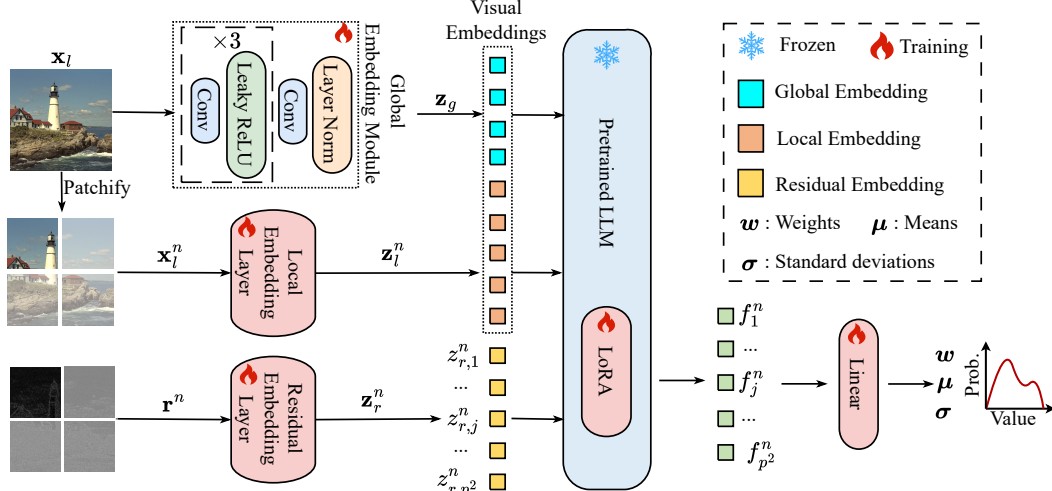

Figure 2: Our distribution estimation framework based on LLM. Visual embeddings, including the global embeddings $\mathbf{z}_g$ and local embeddings $\mathbf{z}_l^n$, enhance the inference. The output feature of LLM $f^n$ are projected onto a Gaussian Mixture Model (GMM) to estimate the residual's probability distribution.

as indexes, with the embedding layer jointly optimized with the entire framework. These global and local embeddings together form visual embeddings, supplying the LLM with both global and local visual information about the image. For compressing the residual patch $\mathbf{r}^n$, the learnable Residual Embedding Layer extracts residual embeddings $\mathbf{z}_r^n \in \mathbb{R}^{p^2 \times d}$. These elements allow us to integrate image information with the LLM's prior knowledge, bridging the gap between image and text tasks, ultimately enhancing compression efficiency.

## 3.2 Distribution Estimation Using Large Language Model

In our proposed framework, we utilize the LLM as a conditional probability estimator, leveraging the lossy reconstruction as visual prompts to predict the probability distribution of the residual image. The estimated distribution is then applied to losslessly encode the residual patch via arithmetic coding.

As illustrated in Fig. 2, the visual embeddings for the LLM consist of global embeddings $\mathbf{z}_g$ and local embeddings $\mathbf{z}_l^n$. The residual is compressed in a pixel-by-pixel manner. For each pixel in the residual patch $\mathbf{r}^n$, we employ an autoregressive approach to estimate its probability distribution. Specifically, to predict the probability distribution of residual pixel $r_j^n$ at position $j$ (where $j = 1, 2, \cdots, p^2$), the visual embeddings, along with previously obtained residual embeddings $\{z_{r,1}^n, \ldots, z_{r,j-1}^n\}$, are concatenated into a sequence and fed into the LLM. The LLM then outputs the corresponding prediction, calculated as follows:

$$f_j^n = F(\mathbf{z}_g, \mathbf{z}_l^n, z_{r,1}^n, \ldots, z_{r,j-1}^n) \tag{1}$$

where $f_j^n \in \mathbb{R}^d$ is the output feature of the LLM for the pixel at position $j$.

To estimate the distribution more accurately, we go beyond directly outputting probabilities and instead predict the parameters of the probability distribution. Specifically, we introduce a Gaussian Mixture Model (GMM) (Cheng et al., 2020) for effective distribution modeling. The parameters of the GMM are derived by linearly projecting the LLM output feature $f_j^n$. These parameters include the weights $\boldsymbol{w}_j^n$, means $\boldsymbol{\mu}_j^n$, and standard deviations $\boldsymbol{\sigma}_j^n$. Consequently, the probability distribution of the residual values can be expressed as follows,

$$p(r_j^n | \mathbf{x}_l, \mathbf{x}_l^n, r_{<j}^n) = p(r_j^n | f_j^n) \sim \sum_{k=1}^{K} \boldsymbol{w}_j^{n,(k)} \mathcal{N}(\boldsymbol{\mu}_j^{n,(k)}, \boldsymbol{\sigma}_j^{2\,n,(k)}) \tag{2}$$

where $k$ denotes the index of mixtures, $K$ denotes the total number of mixtures, and $\mathcal{N}(\mu, \sigma^2)$ denotes a Gaussian distribution with mean $\mu$ and standard deviation $\sigma$.

## 3.3 LOSS FUNCTION

In our proposed method, the primary objective is to minimize the discrepancy between the estimated distribution $p(r)$ and the real distribution $q(r)$. We quantify this discrepancy using cross-entropy: the lower the cross-entropy, the closer $p(r)$ approximates $q(r)$, resulting in fewer bits required by the entropy coder to encode $r$. Specifically, we train our model by optimizing the following loss function:

$$
\begin{aligned}
\mathcal{L} = H(q, p) = \mathbb{E}_{r \sim q}[-\log p(r)] &= -\sum_r q(r) \log p(r) \\
&= -\sum_{n=1}^{N} \sum_{j=1}^{p^2} \log \left\{ \sum_{k=1}^{K} \boldsymbol{w}_j^{n,(k)} \left[ c^{(k)}(r_j^n + \frac{1}{2}) - c^{(k)}(r_j^n - \frac{1}{2}) \right] \right\}
\end{aligned}
\tag{3}
$$

where $c^{(k)}(\cdot)$ is the cumulative distribution function of a Gaussian distribution defined by the mean $\boldsymbol{\mu}_j^{n,(k)}$ and the standard deviation $\boldsymbol{\sigma}_j^{n,(k)}$.

## 4 EXPERIMENTAL RESULTS

### 4.1 EXPERIMENTAL SETTINGS

**Training Details.** We train the entire framework in two stages. In the first stage, we freeze the LLM and optimize all other modules. This stage is trained on the ImageNet2012 dataset (Russakovsky et al., 2015) using the AdamW optimizer (Loshchilov, 2017) with a learning rate of $1 \times 10^{-4}$. In the second stage, we apply the LoRA (Hu et al., 2021) to finetune the LLM. For this, we utilize the DIV2K training dataset (Ignatov et al., 2019) to finetune the entire framework.

In this paper, we use LLaMA3-8B (Dubey et al., 2024) as the default LLM for all experiments. The original images are lossy compressed using BPG (Bellard, 2018) with the compression parameter of $Q = 28$. The lossy reconstructions and the original images are then randomly cropped into patch pairs of size $16 \times 16$, which serve as inputs to the model. To model the distribution, we employ a Gaussian Mixture Model with $K = 5$.

Our method is implemented using the PyTorch framework (Paszke et al., 2017) and requires 3 days to train the entire model on 4 NVIDIA A100 GPUs. Additionally, the arithmetic coding is implemented using the yaecl tool library (Xu et al., 2022).

**Datasets.** To evaluate the performance of the model, we select four different datasets. 1) *DIV2K* (Ignatov et al., 2019): This dataset contains 100 high-resolution color images. 2) *CLIC.mobile* (Toderici et al., 2020): The CLIC mobile validation dataset consists of 61 color images taken with mobile phones, with most images in 2K resolution. 3) *CLIC.pro* (Toderici et al., 2020): The CLIC professional validation dataset includes 41 color images captured by professional photographers, with the majority of images in 2K resolution. 4) *Kodak* (Kodak, 1993): This dataset contains 24 uncompressed $768 \times 512$ color images and is widely used as a benchmark for lossy image compression.

**Baseline Codecs.** To validate the effectiveness of our method, we compare it against eight traditional lossless image encoders: PNG (Boutell, 1997), JPEG-LS (Weinberger et al., 2000), CALIC (Wu & Memon, 1997), JPEG2000 (Skodras et al., 2001), WebP (Google, 2010), BPG (Bellard, 2018), FLIF (Sneyers & Wuille, 2016), and JPEG-XL (Alakuijala et al., 2019). In addition, we include five representative learning-based lossless image compression methods for comparison: L3C (Mentzer et al., 2019), RC (Mentzer et al., 2020), iVPF (Zhang et al., 2021b), iFlow (Zhang et al., 2021a), and DLPR (Bai et al., 2024). We also reproduce the LLM-based lossless image codec (Delétang et al., 2023) in our experiments. Since the LLM used in their approach is not open-source, we substitute it with LLaMA3-8B as the default model while following their other settings.

Table 1: Lossless image compression performance (bpsp) of our proposed method compared to other lossless image codecs on DIV2K, CLIC.pro, CLIC.mobile and Kodak datasets.

| Codec | DIV2K | CLIC.pro | CLIC.mobile | Kodak |
|---|---|---|---|---|
| PNG | 4.23 | 3.93 | 3.93 | 4.35 |
| JPEG-LS | 2.99 | 2.82 | 2.53 | 3.16 |
| CALIC | 3.07 | 2.87 | 2.59 | 3.18 |
| JPEG2000 | 3.12 | 2.93 | 2.71 | 3.19 |
| WebP | 3.11 | 2.90 | 2.73 | 3.18 |
| BPG | 3.28 | 3.08 | 2.84 | 3.38 |
| FLIF | 2.91 | 2.72 | 2.48 | 2.90 |
| JPEG-XL | 2.79 | 2.63 | 2.36 | 2.87 |
| L3C | 3.09 | 2.94 | 2.64 | 3.26 |
| RC | 3.08 | 2.93 | 2.54 | - |
| iVPF | 2.68 | 2.54 | 2.39 | - |
| iFlow | 2.57 | 2.44 | 2.26 | - |
| DLPR | 2.55 | 2.38 | 2.16 | 2.86 |
| Delétang et al. | 4.25 | 3.99 | 4.12 | 4.84 |
| Ours(w/o LoRA) | 2.81 | 2.71 | 2.50 | 3.19 |
| Ours(w/ LoRA) | **2.29** | **2.25** | **2.07** | **2.83** |

**Metric.** We use bits per subpixel (bpsp) as the metric to evaluate the compression ratios. The bpsp is calculated by dividing the total bits in the compressed file by the number of subpixels, where each RGB pixel consists of three subpixels.

## 4.2 MAIN RESULTS

As shown in Table 1, our proposed method achieves state-of-the-art lossless compression performance across all test datasets. On the high-resolution DIV2K and CLIC datasets, our approach further reduces file size by 12.3%-17.9% compared to the best traditional lossless compression scheme JPEG-XL. When compared to SOTA learning-based methods such as DLPR (Bai et al., 2024) and iFlow (Zhang et al., 2021a), our approach also demonstrates superior results. For example, the bpsp of DLPR is 2.55, while our method achieves 2.29, reflecting a 10.2% improvement. Additionally, in comparison with a LLM-based codec (Delétang et al., 2023), our method reduces the bpsp from 4.84 to 2.83 on the Kodak dataset. These results clearly demonstrate that LLMs can be effectively applied to lossless image compression, surpassing even the latest SOTA compression methods. Moreover, these results underscore how our architecture, enhanced with visual prompts, significantly improves the performance of LLM-based codecs in the lossless image compression task.

## 4.3 ABLATION STUDIES

To further analyze our architecture, we conduct ablation studies on the Kodak dataset as shown in Table 2.

**Visual Prompts.** We begin by establishing a simple baseline where the LLM and its original embedding layer are fixed, without the use of visual prompts. Experimental results show that introducing local visual prompts, i.e. the information from lossy patch, reduces the bpsp from 4.84 to 4.09, underscoring the effectiveness of our proposed local visual prompt strategy.

Furthermore, incorporating global visual prompts, i.e. the entire lossy reconstruction, yields an even greater improvement, with a 33.7% reduction in bpsp. Notably, combining both local and global visual prompts leads to further performance gains, demonstrating the effectiveness of visual prompts in enhancing the LLM-based compression framework.

**Learnable Embeddings.** In our proposed framework, the embeddings are optimized jointly with the entire model, rather than utilizing pretrained textual embeddings. To validate the effectiveness

Table 2: Ablation experiments results on the Kodak dataset, using bpsp as the metric.

| Prompt | | Optimized Embeddings | bpsp | Gain |
|---|---|---|---|---|
| Local | Global | | | |
| ✗ | ✗ | ✗ | 4.84 | - |
| ✓ | ✗ | ✗ | 4.09 | -15.5% |
| ✗ | ✓ | ✗ | 3.25 | -32.9% |
| ✓ | ✓ | ✗ | 3.21 | -33.7% |
| ✗ | ✗ | ✓ | 3.40 | -29.8% |
| ✓ | ✓ | ✓ | **3.19** | **-34.1%** |

of this design, we conduct an ablation study by introducing the learnable embedding layer for the baseline approach. As shown in Table 2, the proposed method achieves a bpsp of 3.40, representing a 29.8% reduction compared to the baseline (bpsp = 4.84). This demonstrates that the original LLM embeddings are not well-suited for processing image pixel data and that optimizing the embeddings specifically for image tasks could enhance performance.

**Patch Size.** In our main experiments, we use a patch size of $16 \times 16$ and then extend our evaluation to $24 \times 24$. Increasing the patch size results in a slight performance improvement, with the bpsp decreasing from 3.19 to 3.16. This enhancement can be attributed to the larger patch sizes, which allow for longer contexts that provide more information for the model to process. This additional information enhances the model's ability to capture intricate details and relationships within the image data, ultimately facilitating better compression.

**LLM Size.** We conduct experiments utilizing three LLaMA models with varying parameters and test them on the Kodak dataset to evaluate the impact of LLM size on compression performance. As shown in Table 3, the results indicate that compression performance decreases as the model size decreases; however, the degradation in performance is not significant, as smaller models can still achieve acceptable performance.

Table 3: Results comparison by LLM size.

| Method | bpsp | Loss |
|---|---|---|
| Ours(1B) | 3.24 | 1.6% |
| Ours(3B) | 3.21 | 0.6% |
| Ours(8B) | 3.19 | - |

**Finetuning LLM.** We also investigate various configurations of the LoRA for our proposed LLM-based codec. The specific LoRA settings used for finetuning all linear layers in the LLM are detailed in Table 4. Experimental results indicate that finetuning a limited number of parameters using LoRA leads to significant improvements in lossless image compression performance. However, as the number of finetuning parameters increases beyond a certain point, the compression performance of the finetuned LLM remains essentially unchanged. To strike a balance between compression efficiency and computational cost, we set the rank and alpha to 64 and 128, respectively, in our experiments.

Table 4: Ablation experiments for LoRA, test results on the Kodak dataset, using bpsp as a metric.

| Method | Rank | Alpha | Params | Trainable | bpsp | Gain |
|---|---|---|---|---|---|---|
| w/o LoRA | - | - | - | - | 3.19 | - |
| w/ LoRA | 8 | 16 | 21M | 0.29% | 2.86 | -10.3% |
| | 16 | 32 | 42M | 0.58% | 2.85 | -10.7% |
| | 32 | 64 | 84M | 1.16% | 2.83 | -11.3% |
| | 64 | 128 | 168M | 2.29% | 2.83 | -11.3% |
| | 128 | 256 | 336M | 4.49% | 2.84 | -11.0% |

**BPG Settings.** In this experiment, we evaluate the impact of the quantization parameter (QP) in the BPG codec on the performance of our proposed framework. We train our framework using different QP values, with the corresponding results presented in Table 5. While a lower QP increases the bpsp for lossy compression, it decreases the bpsp for lossless residual compression. Experiments show the final bpsp results are similar in range between [22, 34] and the QP value has a limited influence. Based on these findings, we select BPG with a QP value of 28 as the default lossy codec in our experiment. Notably, our framework is flexible and can incorporate other lossy codecs, such as JPEG, as detailed in the Appendix F.

Table 5: Ablation experiments for QP of BPG, test results on the Kodak dataset, using bpsp as a metric.

| QP | Lossy | Residual | Total |
|----|-------|----------|-------|
| 14 | 0.95 | 2.43 | 3.38 |
| 22 | 0.48 | 2.72 | 3.20 |
| 28 | 0.27 | 2.92 | 3.19 |
| 34 | 0.13 | 3.13 | 3.26 |
| 42 | 0.04 | 3.38 | 3.42 |

### 4.4 LOSSLESS COMPRESSION FOR IMAGES ACROSS DIVERSE DOMAINS

In this section, we apply our proposed pipeline to images from various domains, including screen content images (SCIs) and medical images. Traditional codecs often require specialized tools, such as the intra block copy technique for SCIs, to improve compression performance, which introduces additional design complexity (Xu et al., 2016). In contrast, learning-based codecs can adapt to these diverse image types through training on sufficiently large datasets. Our proposed pipeline further advances by leveraging the extensive prior information embedded in the LLM, resulting in enhanced compression performance across these diverse image types.

**Screen Content Image Compression.** Screen Content Images (SCIs) typically contain text and graphics, with computer-generated elements constituting over 90% of SCIs. Compared to natural images, SCIs are characterized by sharp edges, a limited color palette, high contrast, and markedly different regional complexity, often exhibiting little to no noise (Nguyen et al., 2021).

In this experiment, we utilize HM-SCC (Xu et al., 2016) as the default lossy codec (QP=28). We evaluate performance on the SCID dataset (Ni et al., 2017), with the results presented in Table 6. The results indicate that our method, finetuned on the natural image dataset (i.e., DIV2K), demonstrates competitive generalization ability and can be effectively applied to the SCI domain, achieving a 5.1% improvement over DLPR. Furthermore, finetuning on the SCI dataset DSCIC (Wang et al., 2024) significantly enhances the model's performance within the SCI domain, reaching a SOTA level with a bpsp of 1.11, representing a substantial improvement of 10.5% compared to JPEG-XL.

**Medical Image Compression.** Traditional lossless image compression methods, such as PNG and JPEG-XL, individually encode each slice of 3D medical images. In addition, video coding techniques like HEVC (Sullivan et al., 2012) and VVC (Bross et al., 2021), along with traditional medical image compression method JP3D (Bruylants et al., 2009), treat 3D medical images as video sequences or volumetric data. The latest learned lossless compression methods, including L3C (Mentzer et al., 2019), ICEC (Chen et al., 2022), and aiWave (Xue et al., 2022), are also used as baselines.

Given that medical images are three-dimensional, we split the input medical images into 3-channel slices for processing. In this experiment, we use JPEG-XL as our lossy codec, empirically setting the corresponding quality to 68. Following prior work (Chen et al., 2022), our framework is finetuned on the MRNet training dataset (Bien et al., 2018) and tested on the MRNet validation dataset. The test results are presented in Table 7.

Our model demonstrates superior compression performance for lossless medical image compression. For the Axial subset, the average bpsp of the proposed method is 4.46, compared to 4.72 for JPEG-XL. Moreover, when compared to the learning-based lossless codec L3C (Mentzer et al., 2019), which is also finetuned on medical images in this experiment, our approach shows significantly better compression performance. On the Coronal subset, our method further saves 6.1% bit consumption compared with aiWave (Xue et al., 2022). This improvement can be attributed to our method's utilization of the extensive prior information embedded in LLMs, enhancing overall performance.

Table 6: Applying our model to screen content image compression, test results on the SCID dataset, using bpsp as a metric.

| Codec | bpsp | Gain |
|---|---|---|
| PNG | 1.79 | +14.0% |
| BPG | 1.57 | - |
| WebP | 1.28 | -18.5% |
| JPEG-XL | 1.24 | -21.0% |
| HM-SCC | 1.18 | -24.8% |
| L3C | 2.67 | +70.1% |
| DLPR | 1.58 | +0.6% |
| Ours(DIV2K) | 1.50 | -4.5% |
| Ours(SCI) | **1.11** | **-29.3%** |

Table 7: Applying our model to medical image compression, test results on the MRNet dataset, using bpsp as a metric.

| Codec | Axial | Coronal | Sagittal |
|---|---|---|---|
| PNG | 5.36 | 4.58 | 5.58 |
| JP3D | 4.98 | 4.15 | 5.28 |
| JPEG-XL | 4.72 | 3.89 | 5.09 |
| HEVC | 5.19 | 4.47 | 5.58 |
| VVC | 4.96 | 4.10 | 5.32 |
| L3C | 5.16 | 4.45 | 5.52 |
| ICEC | 4.64 | 3.84 | 4.97 |
| aiWave | 4.55 | 3.80 | 4.83 |
| Ours | **4.46** | **3.57** | **4.83** |

## 4.5 Computational Complexity

Although our LLM-based codec demonstrates superior performance, surpassing classical and other learned-based codecs through its advanced intelligence, its decoding time, as shown in Table 8, is considerably slower than other baselines. This is primarily due to the inherent limitations of autoregressive models and the large number of parameters in LLMs.

Table 8: Comparison of runtimes and kMACs on Kodak dataset.

| Codec | Params | Enc/Dec kMACs/pixel | Enc/Dec Times (s/image) |
|---|---|---|---|
| L3C (Mentzer et al., 2019) | 5M | 252.59/431.31 | 8.17/7.89 |
| DLPR (Bai et al., 2024) | 37M | 18.72/13.37 | 1.26/1.80 |
| Delétang et al. (2023) | 8B | $2.1 \times 10^7$ | 10.44/288.0 |
| Ours (1B) | 1B | $5.9 \times 10^6$ | 3.84/141.6 |
| Ours (3B) | 3B | $1.7 \times 10^7$ | 10.08/338.4 |
| Ours (8B) | 8B | $4.2 \times 10^7$ | 21.12/495.6 |

## 5 Limitations and potential

As illustrated in Figure 1, our approach involves pixel-by-pixel autoregressive coding, which can be time-consuming, although patch-level parallelization is possible. Additionally, by applying acceleration techniques such as distillation and pruning, the inference speed of LLMs is expected to improve, although this aspect falls outside the scope of our current work. Our main objective is to demonstrate the potential of LLMs for lossless image compression and to propose a feasible architecture that leverages the powerful knowledge embedded in LLMs. There are also several methods that could further enhance LLM-based codecs. For instance, replacing BPG (Bellard, 2018) with a learning-based lossy image compression method could enable end-to-end joint optimization, leading to improved results. Additionally, incorporating more sophisticated and efficient context models could help extract richer and more effective features for better compression performance.

## 6 Conclusion

Our work demonstrates that LLMs hold significant potential for lossless image compression. By designing embeddings tailored for image data and incorporating visual prompts, we achieve state-of-the-art lossless compression performance. Additionally, this framework can be effectively adapted to other image compression domains, such as screen content images and medical images. While our exploration of this framework is still in its early stages, we believe that this LLM-based method has the potential to become a new paradigm for image compression in the near future.

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

## A    AUTOREGRESSIVE RESULTS WITH MAXIMUM PROBABILITY

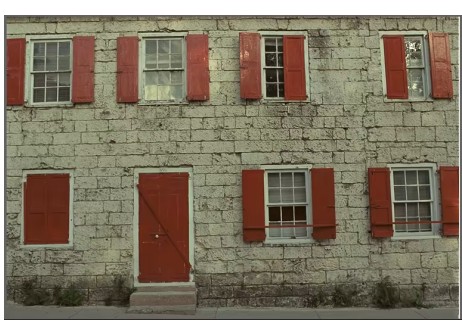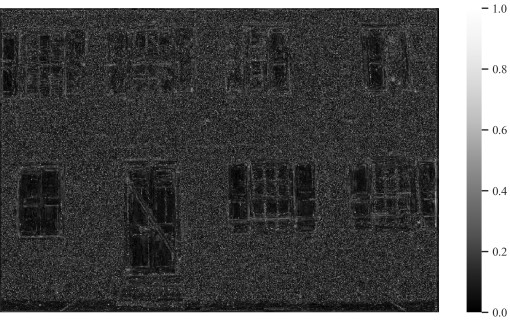

Figure 1: The lossy reconstruction (left) and its corresponding residual image (right) of Kodak 01.

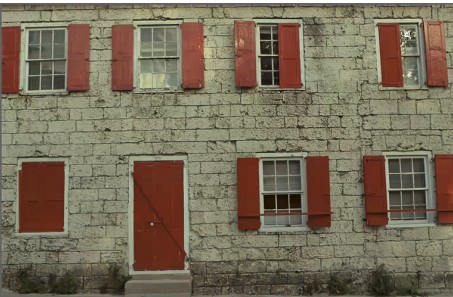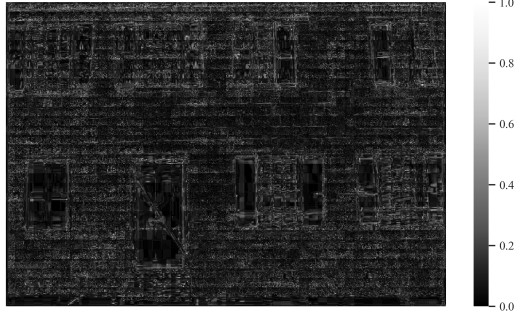

Figure 2: The autoregressive results with only visual prompts (right) and the corresponding reconstructed image (left) of Kodak 01.

Following the research by Delétang et al. (2023), we utilize our compressor as a generative model to visualize the results. The left panel of Fig. 1 displays the outcome of lossy compression applied to the original image using the BPG with QP = 28. In contrast, the right panel illustrates the residuals between the lossy reconstructed image and the original image. To visualize these residuals, we compute their absolute values, average them across the three color channels, and normalize the results to a range between 0 and 1.

Our model employs only visual prompts as input to autoregressively generate the residual values, without any information about the true residuals. For each sampling, we select the residual value with the highest probability, and the final generated residual values are presented in the right panel of Fig. 2. The left panel of Fig. 2 shows the reconstruction results obtained by adding the generated residual values to the BPG lossy image.

In study (Delétang et al., 2023), it is evident that using gzip directly as a generative model results in significant noise, and their method produces incoherent samples. In contrast, our approach, which also functions as a generative model, demonstrates a substantial ability to restore the information from the original image, rather than merely generating meaningless residual values.

## B    BITS ALLOCATION

We visualize the bits allocation results of our method in Fig. 3. Our approach compresses the residual image losslessly using the LLM with visual prompts. As shown in Fig. 3, our method allocates more bits to larger residual values. Although we divide the entire image into small patches, we still provide global and local information through visual prompts. Consequently, more bits are assigned to the high-frequency components of the original image, which are the parts typically lost in lossy coding.

## C    PROBABILITY DISTRIBUTION

A longer context allows LLMs to leverage more sequential dependencies, witch leads to improved performance. In our approach, this translates to achieving better compression efficiency. We visu-

alize the estimated probability distributions of the residuals at different positions within the same patch. These distributions are generated with different amount of available contextual information. Pixels decoded later in the sequence benefit from richer context, witch enables more accurate predictions. As shown in Fig. 4, the estimated probability distributions for later decoding positions are more centralized and precise than those for earlier positions.

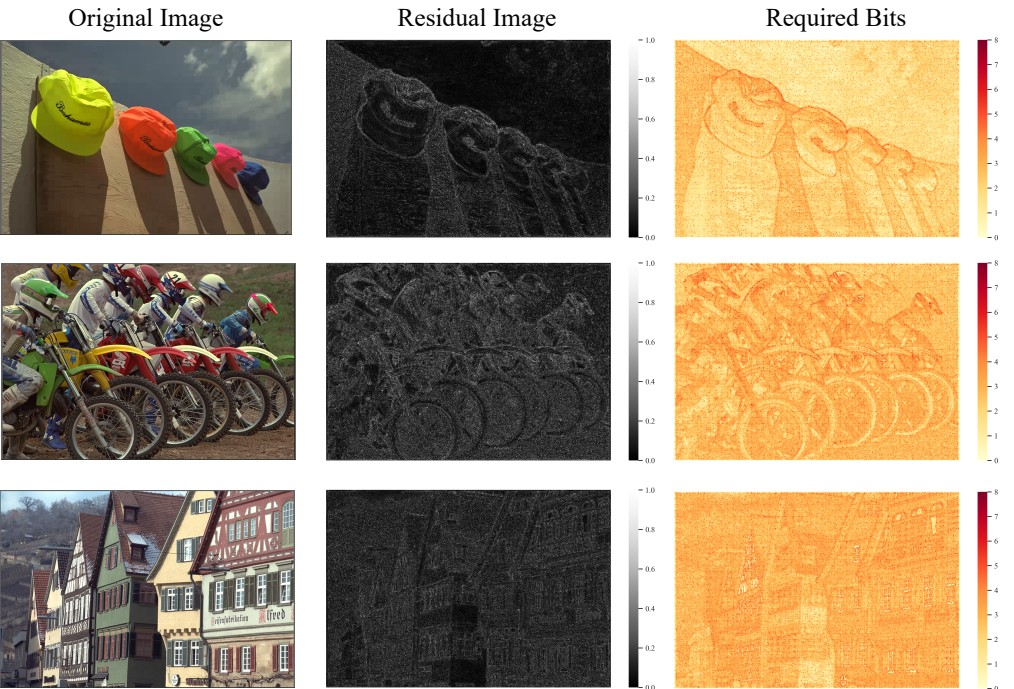

Figure 3: The residual image from BPG lossy compression and the bits allocation using our method. The bpsp is calculated as the mean across the channel dimension.

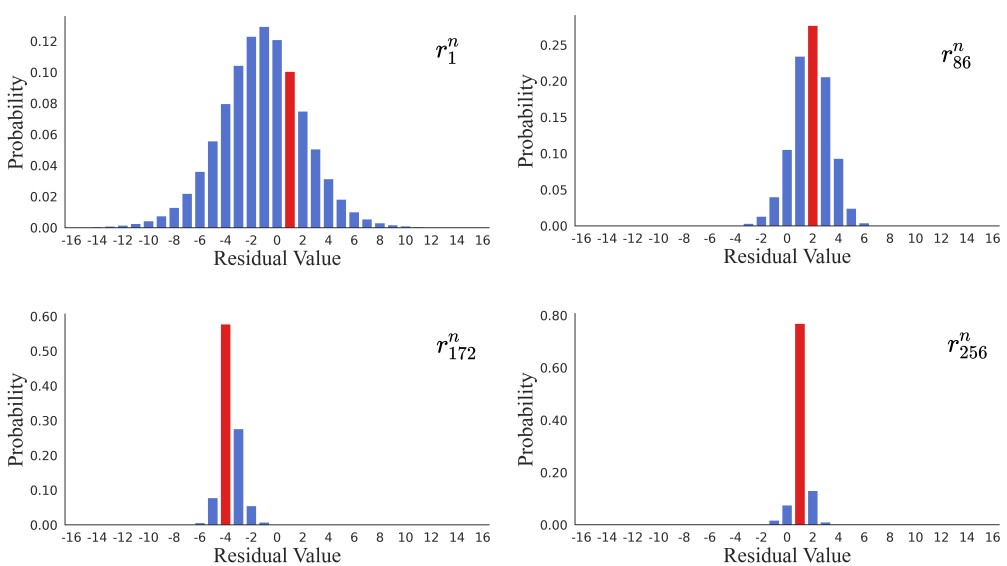

Figure 4: The estimated probability distribution of $r_1^n, r_{86}^n, r_{172}^n, r_{256}^n$ from the same residual patch. The red bar is the real residual value.

# D  COSINE SIMILARITY OF EMBEDDINGS

We visualize the cosine similarity between embeddings in Fig. 5. Pretrained embeddings for text typically exhibit weak correlations between embeddings with neighbour indexes. However, in image domain, neighbouring pixel values are strongly correlated, which have not been exploited by the pretrained embedding layer. Our experiments show that by incorporating the embedding layer into the optimization procedure significantly increases the similarity between neighbouring pixel and residual embeddings, especially for the latter. These results suggest that designing a visually friendly embedding layer for image compression tasks could be advantageous.

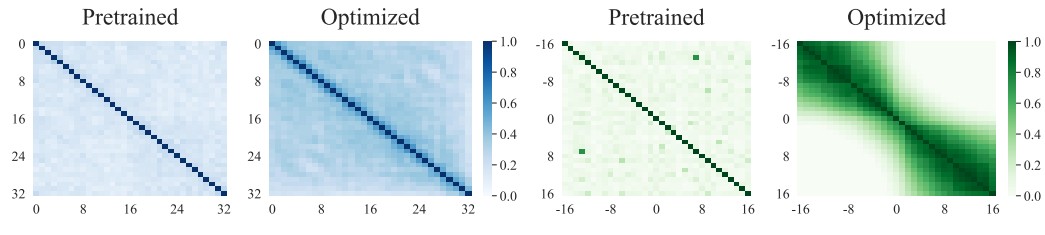

(a) Cosine similarity of local embeddings.    (b) Cosine similarity of residual embeddings.

Figure 5: Cosine similarity between embeddings obtained from pretrained embedding layer and optimized embedding layer.

# E  COMPUTATIONAL COMPLEXITY

In this section, we analyze the complexity introduced by the proposed method and the contribution of each module to the overall computational load.

We separately evaluate the computational overhead of the visual embedding module and the LLM. The results indicate that the additional parameters introduced by the visual prompts module constitute only a small fraction of the total, with a minimal impact on the overall kMACs.

Table 1: Comparison of computational complexity for Visual Embedding and LLM.

| Module | Visual Embedding | LLM |
|---|---|---|
| kMACs/pixel | $1.1 \times 10^3$ | $4.2 \times 10^7$ |
| Params | 4M | 8B |

As our method can be computational and storage heavy, we further investigate quantizing LLaMA3-8B using four methods: BitsAndBytes (Dettmers et al., 2022), GPTQ (Frantar et al., 2022), AWQ (Lin et al., 2024), and BitNet (Ma et al., 2024). The test results on the Kodak dataset are presented in Table 2. Our findings indicate that 8-bit quantization has a negligible impact on performance, while 4-bit quantization (via AWQ) still achieves relatively good results. However, significant performance degradation is observed with 1-bit quantization due to its substantial impact on the model's representation capability. Additionally, directly replacing LLaMA3-8B with BitNet (LLaMA3-8B-Instruct), which is not aligned with our additional trained modules, further exacerbates the performance loss. In future work, we aim to explore advanced quantization techniques to mitigate such performance losses.

Table 2: Comparison of bpsp on the Kodak using different quantization methods for LLaMA3-8B.

| Method | Ours | bnb 8bit | bnb 4bit | GPTQ 4bit | AWQ 4bit | BitNet 1.58bit |
|---|---|---|---|---|---|---|
| **bpsp** | 3.19 | 3.23 | 4.00 | 3.65 | 3.43 | 5.71 |

# F  ABLATION ON LOSSY IMAGE CODEC

We further introduce JPEG as an additional lossy codec. Our findings reveal that, within an appropriate quality range, the lossy codec can be an appropriate component of our architecture. However, extreme quality settings can lead to significant performance degradation. Similar trends are observed between BPG (in Section 4.3) and JPEG; when an appropriate quality is selected, our framework consistently maintains high performance. For example, with BPG, setting the QP value too low reduces performance, as the bitrate required for lossy coding increases significantly. Conversely, setting the QP value too high also degrades performance due to the excessive residuals that must be compressed losslessly.

Table 3: Performance of JPEG as lossy image codec

| Lossy Codec | Lossy | Residual | Total |
|---|---|---|---|
| JPEG (quality=30) | 0.20 | 3.30 | 3.50 |
| JPEG (quality=50) | 0.29 | 2.99 | 3.28 |
| JPEG (quality=70) | 0.40 | 2.96 | 3.36 |

# G  ABLATION ON GMM

GMM is commonly used in image compression(Cheng et al., 2020; Bai et al., 2024). Residual image samples often exhibit complex distributions due to their high-frequency nature, making them challenging to model. Compared to the Gaussian Single Model (GSM), GMM incorporates a minimal increase in parameters while providing significantly improved modeling capabilities. Our ablation study on the number of mixtures K in GMM indicates that $K = 5$ significantly outperforms $K = 1$, highlighting its superior ability to capture complex distributions.

Table 4: Performance comparison for different $K$ Values in GMM

| bpsp | K=1 | K=5 |
|---|---|---|
| Kodak | 3.29 | 3.19 |

# H  ABLATION ON VISUAL PROMPTS

To explore the role of visual prompts in conjunction with LoRA, we conduct the finetuning experiments based on the method of Delétang et al. (2023), and the experimental results are presented in Table 5. It is evident that, after applying the LoRA finetuning, our visual prompts continue to achieve a performance gain of 9.8% to 12.7% on high-resolution DIV2K and CLIC datasets.

Table 5: Performance comparision after applying LoRA for Delétang et al. and Ours.

| Codec | DIV2K | CLIC.pro | CLIC.mobile | Kodak |
|---|---|---|---|---|
| Delétang et al. | 2.54 | 2.50 | 2.34 | 3.00 |
| Ours | **2.29** | **2.25** | **2.07** | **2.83** |

