# OpenReview forum: "Large Language Model for Lossless Image Compression with Visual Prompts"
_ICLR.cc/2025/Conference — Submitted to ICLR 2025_

### Official Review · Reviewer_GpA4 · 2024-11-03

**Soundness:** 2
**Presentation:** 3
**Contribution:** 2
**Rating:** 5
**Confidence:** 3

**Summary:**

This work leverages LLM for lossless image compression. The main contribution is the visual prompts with global and local information for LLM. The lossy reconstruction encoder-decoder is introduced to generate lossy reconstruction embeddings and residual embeddings in residual compression. The proposed method works well on several benchmark datasets and can also be applied to other image domains such as screen content images and medical images.

**Strengths:**

1. The method combines traditional lossy reconstruction and LLM, showing an effective way for lossless image compression.
2. The results look good, the proposed method outperforms all other works across different datasets.

**Weaknesses:**

1. The main contributions of this paper are the visual prompts of lossy reconstruction and the residual compression via LLM, both of which use only existing techniques (e.g., visual prompts and LoRA) and thus the novelty is limited. Moreover, I didn't see the "visual prompts" module in all figures, please clarify it in at least one of these figures.
2. What is the arithmetic coding? The authors should provide some background on it (or at least add it somewhere in the appendix). Otherwise it would be very confusing for people who haven't worked in this area.
3. The proposed method requires an additional lossy encoder-decoder in compression, which increases the training cost and also lowers the inference time. The authors should provide a comparison of model complexity and training/inference time.
4. How do you design the Residual embedding layer? Pixel values as indexes or other ways? Please clarify it in Sec.3.1, Line 250-251. In addition, what's the mean of "use pixel values as indexes", is that an image histogram embedding?
5. Although this method introduces the spatial information as a part of embeddings for LLM, the output is still a 1-D distribution. Maybe the authors can extend it to 2-D spatial distribution and optimize it with 2D entropy?
6. By introducing BPG as the lossy reconstructor, this method seems to have an additional compression vector (i.e., larger bits than other models). The experiments might be unfair. The authors should explain more about it.

**Questions:**

1. In Figure 1, the existing LLM-based pipeline doesn't have trainable parameters. It seems strange because most visual embeddings are not naturally aligned with text embeddings in LLMs.
2. In Table 1, the method proposed by Deletang et al. performs the worst. Why?
3. In Table 2, the Global prompt seems to be the most effective component. Can the authors also show the cosine similarity of global embeddings in Figure 4?

---

> ### Author Response · Authors · 2024-11-22
> **Response to reviewer GpA4[1/2]**
>
> Thank you for your insightful questions. We have prepared detailed, point-by-point responses to each query. We hope this addresses your concerns effectively.
>
> **Q1: About Visual Prompt Module**
>
> **A1:** Thanks for your question. Our objective is to showcase the feasibility and significant potential of LLM-based models for image compression. In our approach, a lossy codec is utilized to generate visual prompts, while residuals are compressed using LLMs enhanced by these prompts. By integrating relatively simple modules, we achieve state-of-the-art performance, underscoring the advantages of LLM-based approaches. In contrast, existing neural network-based methods increasingly rely on complex architectures, leaving limited room for further significant improvements.
>
> In our method, "visual prompts" specifically refer to the lossy image outputs processed through global and local modules to produce visual embeddings. To enhance clarity, we will update the figure in the final version with additional annotations.
>
> **Q2: About arithmetic coding**
>
> **A2:** Thanks for your question. We apologize for the oversight and will include a more detailed background in the related work. Giving probability distribution and input sequence, arithmetic coding is a lossless data compression technique that generates nearly optimal-length codes. It encodes an entire message as a single number within the interval \[0, 1\) (represented in binary), using a probabilistic model to subdivide the interval into subintervals proportional to each symbol's probability. As symbols are sequentially processed, the interval is refined, and the final number within it uniquely represents the complete message. Widely used in image and video compression, arithmetic coding losslessly reduces bitrates.
>
> **Q3: About model complexity**
>
> **A3:** Thanks for your question. We separately calculate the computational overhead introduced by the visual embedding module and the LLM. The results show that the additional parameters from the visual prompts module account for only a small fraction of the total and have a negligible impact on the overall kMACs.
>
> | Module      | Visual Embedding | LLM             |
> | ----------- | ---------------- | --------------- |
> | kMACs/pixel | $1.1\times10^3$  | $4.2\times10^7$ |
> | Params      | 4M               | 8B              |
>
> We use 8 NVIDIA A100 GPUs to measure the time taken to encode one image (Kodak dataset, 768x512 resolution). We provide the per-pixel kMACs for the main comparison methods and observe that our approach significantly reduces kMACs with minimal performance degradation when employing a smaller LLM (1B/3B).
>
> | Kodak           | kMACs/pixel     | Enc/Dec Time(second) | bpsp |
> | --------------- | --------------- | -------------------- | ---- |
> | Deletang et al. | $2.1\times10^7$ | 10.44/288.0          | 4.84 |
> | Ours(1B)        | $5.9\times10^6$ | 3.84/141.6           | 3.24 |
> | Ours(3B)        | $1.7\times10^7$ | 10.08/338.4          | 3.21 |
> | Ours(8B)        | $4.2\times10^7$ | 21.12/495.6          | 3.19 |
>
> **Q4: About embedding layer**
>
> **A4:** Thanks for your question. Our local embedding layer and residual embedding layer constitute the embedding module, which functions essentially as a lookup table. Each index in this table corresponds to a vector. For the local layer, the pixel value serves as the index to retrieve its corresponding vector, with pixel values ranging from [0, 255]. In the case of the residual layer, the residual value is used as the index to access the corresponding vector through embedding. Since the residual values span [-255, 255], an offset is applied to adjust the range to [256, 767], ensuring the indexes align appropriately with the embedding vectors.
>
> **Q5: About 2D entropy**
>
> **A5:** Thanks for your suggestion. Our current approach predicts a unique probability distribution for each residual value and leverages it for compressing the residuals, a method widely adopted in learning-based image and video compression. While 2D entropy is beyond the scope of this paper, we adhere to the commonly used 1D entropy modeling for a fair comparison.
>
> **Q6: About BPG**
>
> **A6:** Thanks for your question. In our experiments, the total stream comprises two parts: the stream of lossy images encoded by the BPG and the stream of residuals compressed using the LLM-based model. The contribution of the BPG stream has been included in the overall total stream. The impact of the BPG on the entire compression paradigm is illustrated in Table 4.

---

> > ### Author Response · Authors · 2024-11-22
> > **Response to reviewer GpA4[2/2]**
> >
> > **Q7: About the existing LLM-based pipeline in Figure 1**
> >
> > **A7:** Thanks for your question. The existing LLM-based pipeline specifically refers to the method outlined in [1], which directly employs an LLM for image data compression without training on the image data. As a result, it relies on text embeddings without optimizing them for visual data. This creates a mismatch between visual and textual embeddings, leading to suboptimal compression performance. Our work addresses this issue by bridging the gap, ensuring that the embeddings are better suited for image data and improving overall compression efficiency.
> >
> > **Q8: About result of [1] in Table 1**
> >
> > **A8:** Thanks for your question. The method described in [1] has no trainable parameters, and its direct flattening of pixels into one-dimensional sequences disrupts the spatial relationships between them. Therefore, it overlooks the potential to align the visual embeddings and textual embeddings to enhance the compression ability of LLM, which is exactly the problem that our paper wants to solve. Despite these limitations, it achieves performance comparable to PNG, demonstrating the significant potential of the LLM-based modeling paradigm in image compression.
> >
> > [1] Gregoire Deletang et al., Language modeling is compression. In ICLR, 2024
> >
> > **Q9: About the cosine similarity of global embeddings**
> >
> > **A9:** Thanks for your question. For the global embedding, we extract the corresponding features directly using a CNN network, which serves as the global embedding layer in our approach. Unlike local or residual embeddings, we do not employ an explicit embedding lookup table for the global embedding. Consequently, it is not feasible to directly compute the corresponding cosine similarity.

---

> > > ### Comment · Reviewer_GpA4 · 2024-11-25
> > > **Additional comments from the rebuttal**
> > >
> > > I appreciate the author's rebuttal and new experiments on model complexity. Some of my concerns (such as the arithmetic coding and embedding layer) are addressed. However, I am still confused about why using more quantization parameters increases the bpsp for residual compression (in Table 4). If the method of [1] is nontrainable, the comparison in Table 1 should add at least one additional result that finetunes LLM for image compression, as a baseline of the proposed approach.
> > >
> > > More importantly, please revise the draft (and re-upload it) accordingly before the discussion deadline, so that all reviewers and AC can see your updates and re-evaluate the paper.

---

> > > > ### Author Response · Authors · 2024-11-26
> > > >
> > > > Thanks for your response. We're pleased to hear that our previous explanations have addressed your previous concerns, and we're happy to address any additional questions you may have.
> > > >
> > > > Firstly, the quantization parameter (QP) of BPG significantly influences the quality of lossy compressed images. Bigger QP, worse quality. As the QP increases, the quantization step (Qstep) becomes larger, leading to a reduction in the bpsp of the lossy images. However, this also results in increased error, making the residual more complex and requiring higher bpsp for effective compression of the residual.
> > > >
> > > > Secondly, we conduct fine-tuning experiments based on [1] and evaluate the results on the DIV2K dataset, with the outcomes presented in the table below. The experimental results show that, both without LoRA, our visual prompts reduce the bpsp from 4.25 to 2.81, yielding a 34.1% gain. Our method (with visual prompts and LoRA) further reduces the bpsp of [1] (with LoRA) from 2.54 to 2.29, resulting in a 9.8% gain.
> > > >
> > > >
> > > > | DIV2K | [1] w/o LoRA | Ours(w/o LoRA) | [1] w/ LoRA | Ours(w/ LoRA) |
> > > > | ----- | ------------ | -------------- | ----------- | ------------- |
> > > > | bpsp  | 4.25         | 2.81           | 2.54        | 2.29          |
> > > >
> > > >
> > > > Thirdly, our paper is still undergoing refinement, and we will update it within a day/before deadline.
> > > >
> > > > Thank you again for your engagement.
> > > >
> > > > [1] Gregoire Deletang et al., Language modeling is compression. In ICLR, 2024

---

> > > > > ### Author Response · Authors · 2024-11-28
> > > > >
> > > > > Thanks for your suggestions. We have re-uploaded the revised paper, which incorporates additional experiments and suggestions made during the discussion. The revisions are summarized below:
> > > > > * An introduction on arithmetic coding has been added to Section 2.2 (lines 116-121).
> > > > > * Experimental results for [1] and our method (without LoRA) have been included (lines 342-343 and 952-965).
> > > > > * We have moderated our claims regarding the role of optimized embeddings (lines 395-397).
> > > > > * Ablation experiments on LLM size and patch size have been added to Section 4.3 (lines 398-412).
> > > > > * Discussions related to model complexity have been added in Section 4.5 and Appendix E (lines 502-518 and 888-917).
> > > > > * Ablation experiments on the lossy codec (lines 918-935) and GMM (lines 937-950) have also been added to the Appendix.
> > > > >
> > > > > [1] Gregoire Deletang et al., Language modeling is compression. In ICLR, 2024

---

> > > > > > ### Author Response · Authors · 2024-12-03
> > > > > >
> > > > > > Thank you for taking the time to discuss the paper with us. We sincerely hope that we have addressed your concerns. Please let us know if you have any further questions or if there is anything else we can clarify.

---

### Official Review · Reviewer_htWV · 2024-11-03

**Soundness:** 2
**Presentation:** 2
**Contribution:** 2
**Rating:** 3
**Confidence:** 5

**Summary:**

This paper introduces a LLM-based entropy model for lossless image compression. Specifically, it adopts the notion of scalable coding to encode an input image into a base layer and an enhancement layer. The base layer is coded in a lossy manner using BPG while the enhancement (residual) layer is coded losslessly with the proposed LLM-based model. Its key contribution lie in learning visual embeddings tailored for image compression.

**Strengths:**

(1) The idea of using LLM for entropy coding is novel and presents a new research direction.

(2) The ablation studies are extensive and complete, showing the potential of LLM for lossless image compression in various domains, including natural, screen, and medical image content.

**Weaknesses:**

(1) It is unclear how much gain the lossy base layer contributes to the final coding performance, although the authors argue with the results in Table 4 that the base layer has a limited impact. But, I doubt this is true. What if there is no base layer at all?

(2) Obviously, querying a LLM is not cheap. The decoding time and MAC/pixel are not presented.

(3) If the LLM is not fine-tuned (i.e. LoRA is not applied), the proposed method (bpsp = 3.19) performs only comparably to JPEG 2000 on the Kodak dataset (and worse than some traditional codecs). Turing the LLM into a specific entropy coding model does not sound practical.

**Questions:**

(1) In Table 4, I wonder how the proposed method would work if there is NO base layer at all.

(2) From Table 2, it appears that the major coding gain of the proposed method comes from providing the LLM with local and global prompts. In the variant with "disabled optimized embeddings" (gain=-33.7%), I wonder if the visual embeddings z_g and z_l still need to be learned and optimized. Moreover, from the same table, additionally optimizing embeddings offers rather limited gain. This appears to contradict the claim that learning better embeddings for the image compression task is essential. It is unclear what embeddings are additionally optimized. Does the setting "Optimized Embeddings" refer solely to the "residual embedding layer"? This needs to be clarified.

(3) In Table 2, it is counterintuitive that global prompts bring more gain than local prompts. Local prompts should ideally capture better local statistics.

(4) In Figure 1, the proposed method has two types of embedding layers: the textual and visual embeddings. What are the differences between these two. Does the textual embedding refer to the embedding needed to convert a residual sample into z_r? This part is rather ambiguous.

(5) Since the input image encoded by the LLM is a residual image, the samples of a residual image usually have an uni-modal distribution. The necessity of GMM is questionable.

(6) The decoding time (or even the kMAC/pixel) of the proposed method should be compared with that of the competing baselines.

(7) In Table 1, the results without LoRA should be included. Applying LoRA to fine-tune LLM would turn the LLM into a specific entropy coding model for coding images. In Deletang et al, the LLM does not appear to be fine-tuned. Fine-tuning LLM to make it a specific one for image coding does not sound practical.

---

> ### Author Response · Authors · 2024-11-23
> **Response to reviewer htWV[1/3]**
>
> Thank you for your insightful questions. We have prepared detailed, point-by-point responses to each query. We hope this addresses your concerns effectively.
>
> **Q1: About lossy base layer**
>
> **A1:** Thank you for your question. The lossy base layer is a critical component of our design, acting as the source of visual prompts. Within an appropriate QP range, the choice of lossy codec has a limited impact on overall performance. However, unreasonable QP settings can significantly degrade performance. Importantly, this layer is flexible and can be replaced with different codecs or QP settings, underscoring the adaptability and robustness of our overall framework.
>
> If the lossy base codec is removed from our design, the approach would resemble the method proposed in [1]. However, by introducing the lossy base codec to generate visual prompts, our design achieves a significant performance improvement over [1].
>
> We conduct experiments by expanding the range of QP values for BPG and introducing JPEG as an alternative lossy codec. The results show that when QP values are within the range of [22, 34], the bpsp remains relatively stable. However, setting the QP too low (e.g., QP=14) negatively impacts performance due to a significant increase in the bitrate required for lossy coding. Conversely, setting the QP too high (e.g., QP=42) also degrades performance because the excessive residuals demand more resources for lossless compression. Similar trends are observed for JPEG; when an appropriate quality is selected, our framework consistently maintains high performance. Therefore, selecting an appropriate QP range for the chosen lossy codec is essential to maintain optimal performance.
>
> | Lossy Codec      | Lossy | Residual | Total | Compression Ratio |
> | ---------------- | ----- | -------- | ----- | ----------------- |
> | BPG(QP=14)       | 0.95  | 2.43     | 3.38  | 42.2%             |
> | BPG(QP=22)       | 0.48  | 2.72     | 3.20  | 40.0%             |
> | BPG(QP=28)       | 0.27  | 2.92     | 3.19  | 39.8%             |
> | BPG(QP=34)       | 0.13  | 3.13     | 3.26  | 40.7%             |
> | BPG(QP=42)       | 0.04  | 3.38     | 3.42  | 42.7%             |
> | JPEG(quality=30) | 0.20  | 3.30     | 3.50  | 43.7%             |
> | JPEG(quality=50) | 0.29  | 2.99     | 3.28  | 41.0%             |
> | JPEG(quality=70) | 0.40  | 2.96     | 3.36  | 42.0%             |
>
> [1] Gregoire Deletang et al., Language modeling is compression. In ICLR, 2024
>
>
> **Q2: About decoding time and kMACs**
>
> **A2:** Thanks for your question. We acknowledge that LLMs introduce considerable computational complexity; however, this can be significantly mitigated by reducing the number of parameters.
>
> We use 8 NVIDIA A100 GPUs to measure the time taken to encode one image (Kodak dataset, 768x512 resolution). The per-pixel kMACs and time consumed of our method are presented in the table below. Our experiments with LLMs of varying sizes confirm that the proposed method achieves good results even with smaller LLMs. In the future, techniques such as lightweight model architectures, pruning, and quantization can be explored to reduce computational complexity without significant performance degradation.
>
> | Kodak           | Params | bpsp | Enc/Dec Time (s) |
> | --------------- | ------ | ---- | ---------------- |
> | Deletang et al. | 8B     | 4.84 | 10.44/288.0      |
> | Ours(1B)        | 1B+4M  | 3.24 | 3.84/141.6       |
> | Ours(3B)        | 3B+4M  | 3.21 | 10.08/338.4      |
> | Ours(8B)        | 8B+4M  | 3.19 | 21.12/495.6      |
>
> **Q3: About Fine-tuning LLM**
>
> **A3:** Thanks for your question. While LLMs are pre-trained exclusively on language tasks without exposure to image data, our approach demonstrates that they can achieve performance comparable to image-specific codecs(JPEG2000) through the use of visual prompts. These results are significant, as they not only showcase the potential of pre-trained LLMs for image-related tasks but also deepen our understanding of their capabilities. Given that large models are likely to become universal tools in the future, our method requires only 21M parameters for LoRA fine-tuning, enabling LLMs to function as image codecs through a simple plug-in mechanism. This approach presents a promising and practical solution for adapting large models to diverse applications.

---

> > ### Author Response · Authors · 2024-11-23
> > **Response to reviewer htWV[2/3]**
> >
> > **Q4: About effect of optimized embeddings**
> >
> > **A4:** Thanks for your question. In the variant with "disabled optimized embeddings", global embeddings $z_g$ is learned and optimized, while local embeddings $z_l$ and residual embeddings $z_r$ rely on the embedding layer from textual pretraining. This distinct arises because the process of obtaining $z_g$ differs from the general embedding approach----a lookup table. Instead of using a predefined table, $z_g$ is directly derived through convolution.
> >
> > "Optimizing Embeddings" refers to optimizing both the local embedding layer and the residual embedding layer. Consequently, the prior comparison may not have been entirely fair, as it did not fully reflect the contributions of embeddings. To better evaluate the role of embeddings, we conducted additional experiments by removing the global module and optimizing only the local embeddings, resulting in a performance improvement from 15.5% to 32.0%.
> >
> > | Local    | Local(Optimized) | bpsp | Gain |
> > | -------- | -------------------------- | ---- | ---- |
> > | &#10004;         |                            | 4.09 | -15.5%  |
> > | &#10004; |     &#10004;                  | 3.29   | -32.0% |
> >
> > These results clearly highlight the importance of optimized embeddings in enhancing performance. We apologize for any confusion caused by the existing results and will include these additional experiments in the final version for a more comprehensive and balanced comparison.
> >
> >
> > **Q5: About local embedding vs. global embedding**
> >
> > **A5:** Thanks for your question. The local prompt in Table 2, when directly fed with the embedding trained for text by the LLM, achieves only a 15.5% improvement. However, when the embedding is specifically trained for visual tasks, the improvement reaches 32.0%, which is close to the gain achieved by the Global Prompt. The global prompts provides global information while the local prompts supplies local information. However, it is important to note that the input also includes some local information provided by the residuals, which is an inseparable part of our design. We can not further conduct experiments using only global information. This additional local information enhances the global prompts, making it a more effective experiment than the local prompts alone. We will refine the experimental section to include these results and clarify the details in the final version of the paper.
> >
> >
> >
> > | Local    | Local(Optimized) | Global(Optimized) | bpsp | Gain   |
> > | -------- | ---------------- | ----------------- | ---- | ------ |
> > | &#10004; |                  |                   | 4.09 | -15.5% |
> > | &#10004; | &#10004;         |                   | 3.29 | -32.0% |
> > |          |                  | &#10004;          | 3.25 | -32.9% |
> >
> >
> >
> > **Q6: About textual and visual embeddings in Fig 1**
> >
> > **A6:** Thanks for your question. Our Fig. 1 may be confusing, as it serves as a schematic diagram primarily highlighting our contributions: the introduction of visual prompts and the training of the embedding. Specifically, existing schemes [1] only utilize the original textual embeddings. In contrast, we not only use the original textual embeddings but also introduce visual embeddings.
> >
> > Your understanding is correct; in our approach, the residual embedding layer is derived from the original textual embedding layer, while the local embedding layer is also transformed from the same original textual embedding layer. The actual details are illustrated in Figures 2 and 3. Since both global and local embeddings are used as prompts, we refer to them collectively as visual prompts. We will revise Figure 1 in the final version to enhance clarity.

---

> > ### Comment · Reviewer_htWV · 2024-11-26
> > **Thank your for your responses**
> >
> > I like to thank the authors for making effort to address my comments. By looking at the results, I am NOT fully convinced that this is a right direction to go. The reasons are three-fold. First, it is shown that the base-layer quality does have an impact on the overall coding performance. A proper QP must be chosen for the base layer. This proper QP may be image dependent. Second, the encoding/decoding time is still way too high, as expected. Although the idea of using LLM for lossless compression is interesting, its practicality is questionable. Third, although the system requires fine tuning only 21M of parameters, to put the system to use, these 21 M parameters need to be integrated into the LLM for inference, which makes this LLM a task-specific huge network. IMHO, it does not make sense from the application perspective.

---

> > > ### Author Response · Authors · 2024-11-26
> > >
> > > Thank you for your reply.
> > >
> > > First, the QP setting for our base layer within the range of [22, 34] has minimal impact on performance. This range is broad, not image-dependent, and aligns with the reasonable settings for most image compression tasks. In the future, the base layer could be replaced with a learnable lossy compression module, enabling end-to-end training without the need for manual adjustments. However, this is beyond the scope of our current work, as we prioritize a simpler architecture.
> > >
> > > Second, we acknowledge that our solution may not currently prioritize practicality; instead, our focus lies in its academic value. The widespread success of LLMs across various fields promopts us to explore their potential contributions to image compression tasks. We demonstrate that LLMs can easily achieve state-of-the-art performance, indicating substantial unapped potential in LLM-based architectures. We believe this direction holds significant academic value for advancing image compression. Our work aims to inspire more researchers to engage in this field, driving improvements in performance of LLM-based codec, reducing complexity, and ultimately making LLMs practical image codecs.
> > >
> > > We hope our responses address your concerns and help convince you to reconsider your decision. We look forward to hearing your feedback.

---

> > > > ### Author Response · Authors · 2024-12-03
> > > >
> > > > Thank you for your reply. We sincerely hope that our statement on the LLM-based image compression method has convinced you that it is a promising direction. If you have any further questions or need additional clarification, please let us know. We would be glad to provide any information you need!

---

> ### Author Response · Authors · 2024-11-23
> **Response to reviewer htWV[3/3]**
>
> **Q7: About GMM**
>
> **A7:** Thank you for the question. GMM is commonly used in image compression, such as [2][3]. Residual image samples often exhibit complex distributions due to their high-frequency nature, making them challenging to model. Compared to the Gaussian Single Model (GSM), GMM incorporates a minimal increase in parameters while providing significantly improved modeling capabilities. Our ablation study on the number of mixtures K in GMM indicates that K=5 significantly outperforms K=1, highlighting its superior ability to capture complex distributions effect.
>
> | bpsp  | K=1  | K=5  |
> | ----- | ---- | ---- |
> | Kodak | 3.29 | 3.19 |
>
> [2] Zhengxue Cheng et al., Learned Image Compression With Discretized Gaussian Mixture Likelihoods and Attention Modules. In CVPR, 2020
>
> [3] Yuanchao Bai et al., Deep lossy plus residual coding for lossless and near-lossless image compression. IEEE Transactions on Pattern Analysis and Machine Intelligence, 2024
>
> **Q8: About the results without LoRA**
>
> **A8:** We will incorporate the results of the tests conducted without LoRA into Table 1, as shown below. These findings clearly demonstrate that the method proposed in this paper consistently surpasses the approach in [1], achieving a significant performance margin even without the use of LoRA.
>
>
> | bpsp           | DIV2K | CLIC.pro | CLIC.mobile | Kodak |
> | -------------- | ----- | -------- | ----------- | ----- |
> | [1]            | 4.25  | 3.99     | 4.12        | 4.84  |
> | Ours(w/o LoRA) | 2.81  | 2.71     | 2.50        | 3.19  |

---

### Official Review · Reviewer_RowH · 2024-11-04

**Soundness:** 3
**Presentation:** 2
**Contribution:** 2
**Rating:** 5
**Confidence:** 4

**Summary:**

This paper presents an approach to lossless image compression by integrating Large Language Models (LLMs) with visual prompts to enhance compression performance. Traditional lossless compression techniques, also those based on deep learning, struggle to utilize the rich prior knowledge in LLMs, which is predominantly textual. This study mitigates this limitation by introducing a framework that combines lossy image reconstructions as visual prompts to the LLM, which are used to extract local and global visual features. These features, alongside the residuals between the original and lossy images, guide the LLM in predicting probability distributions for the residuals, effectively functioning as an entropy model.

**Strengths:**

- The research direction *LLM-based image compression* is interesting.
- From the experimental results, the proposed method did achieve SOTA compression performance.

**Weaknesses:**

- Although the proposed method demonstrates improved performance, its design elements appear largely incremental, limiting the overall novelty. From the paper’s descriptions, the main factor contributing to the superior results over [1] seems to be the retraining of modules within the LLM, whereas [1] uses these modules in their pretrained state.
- Following this, it remains unclear what the primary differences are between this method and that of [1]. Although the authors briefly mention where [1] differs, they do not provide enough detail for readers to clearly distinguish their approach from prior work.
- Regarding Figure 2, how does the Decoder process unfold? Couldn’t we use the bits encoded by the Arithmetic Encoder directly, then decode them using the Arithmetic Decoder, and simply add the decoded residual to the lossy reconstruction to obtain the final image? It’s unclear why the lossy image and patches need to be passed back through the LLM in the Decoder. Additionally, the purpose of the dotted arrows in the figure could be clarified, as the visual explanation is somewhat confusing.
- How well is the proposed method expected to generalize across different datasets? For instance, if the LLM is fine-tuned on dataset A, how does it perform when applied to compress images from dataset B?
- Why was a large language model (LLM) chosen instead of a vision-language model? Given the fundamental differences between text and image modalities, it seems questionable to use an LLM trained solely on text data to handle image compression.

**Questions:**

Please refer to the weakness part above.

---

> ### Author Response · Authors · 2024-11-22
> **Response to reviewer RowH**
>
> Thank you for your insightful questions. We have prepared detailed, point-by-point responses to each query. We hope this addresses your concerns effectively.
>
> **Q1: About overall contribution && differences from [1]**
>
> **A1:** Thanks for your question. Our primary goal is to demonstrate the potential of LLMs in image compression tasks. Unlike [1], which overlooks the correlations between pixels and the prompt capabilities of LLMs, our method enhances performance by introducing visual prompts derived from a lossy codec. This adjustment significantly boosts the system's overall performance, resulting in a 34.1% improvement. Furthermore, our ablation studies in Tables 2 and 3 highlight the critical role of fine-tuning LLMs with LoRA, which yields an additional 11.3% performance gain. These results underscore the necessity of fine-tuning with LoRA to achieve optimal performance.
>
> [1] Gregoire Deletang et al., Language modeling is compression. In ICLR, 2024
>
> **Q2: About the decoding procedure**
>
> **A2:** Thanks for your question. During the decoding process, arithmetic decoding relies on the corresponding probability distribution to convert the binary stream back into residual values. In our proposed method, this probability distribution is derived using the lossy encoded image. Consequently, the lossy image must be obtained prior to arithmetic decoding, serving as a visual prompt for predicting the probability distribution. The previously decoded residuals act as additional prompts, enabling the LLM to autoregressively compute the distribution for the next residual value to decode.
>
> Therefore, the arithmetic coding process you mentioned cannot be used directly. Instead, the decoded residual values are iteratively fed back into the LLM, as illustrated by the dotted line in Fig. 2. We will clarify this process further in the final version of the paper.
>
>
> **Q3: About training and test set**
>
> **A3:** Thanks for your question. We trained our model using the ImageNet2012 and the DIV2K training datasets and evaluated it on Kodak, DIV2K test dataset, CLIC.mobile, and CLIC.pro, ensuring the training and test set were completely different. Additionally, to validate our method, we further conduct experiments on a larger ImageNet2012 validation set, and for time reasons we only tested the first 200 images, achieving lossless compression results for 2.67 bpsp.
>
>
> | Codec | JPEG-XL | L3C  | DLPR | Ours     |
> | ----- | ------- | ---- | ---- | -------- |
> | bpsp  | 2.91    | 3.30 | 2.77 | **2.67** |
>
>
> **Q4: About vision-language model**
>
> **A4:** Thanks for your question. While a visual language model can be utilized for image tasks, our aim is to demonstrate that an LLM, pre-trained for linguistic tasks, can also be effective for image compression. This represents a new paradigm for image compression tasks. Despite the LLM having no prior exposure to image data, we have successfully employed it to achieve state-of-the-art results. Our approach may pave the way for enhanced performance with visual language models in scenarios where the effectiveness of this new paradigm has already been established.

---

> > ### Comment · Reviewer_RowH · 2024-11-25
> > **Thanks for the authors' response**
> >
> > I have carefully reviewed the authors' response and the comments from other reviewers. I appreciate the authors' efforts in addressing my questions and apologize for overlooking the explicit reference to "[1]" in my initial review.
> >
> > That said, my concerns about the work's contribution remain unresolved. While this paper demonstrates improved performance by retraining LLM modules for image compression tasks, the foundational idea of leveraging LLMs for such tasks was already introduced in Delétang et al.'s work. As a result, I find it difficult to regard the presented improvement as a significant contribution.
> >
> > The authors assert that their primary contribution lies in further exploring the correlation between image pixels and text prompts. However, this correlation is a well-established topic in the vision-language model (VLM) domain. As I pointed out in my initial review, it raises the question of why the authors did not utilize existing VLMs, which are inherently designed to handle such correlations, instead of adapting a raw LLM. From this perspective, the approach of integrating visual tokens into LLMs feels akin to developing yet another vision-language model, which diminishes the novelty of the contribution.
> >
> > Given these remaining concerns, I am inclined to maintain my initial rating.

---

> > > ### Author Response · Authors · 2024-11-26
> > >
> > > Thanks for your response and suggestion.
> > >
> > > Firstly, while [1] explored the use of LLMs for image compression, their work primarily demonstrated the feasibility of this approach, achieving performance comparable to PNG. In contrast, our method introduces visual prompts through lossy image codecs, significantly enhancing performance to state-of-the-art levels. This not only unlocks the greater potential of LLMs in the image compression domain but also surpasses existing methods, representing a significant contribution of our work.
> > >
> > >
> > > Secondly, although the correlation between image pixels and textual prompts has been investigated in the vision-language model (VLM) domain, no related research has addressed this issue in the context of lossless image compression tasks, and an effective solution remain unclear. Our approach offers a viable solution to enhance the performance of lossless image compression by utilizing lossy images as visual prompts.
> > >
> > >
> > > Finally, it is important to note that image compression, as a low-level task, requires preserving fine details as much as possible. However, existing VLM works primarily focus on high-level tasks, often capturing high-level semantics while failing to retain low-level details. Directly using VLMs for tasks such as image reconstruction may produce images with highly correlated semantics but differing details. To the best of our knowledge, VLM-based methods have yet to demonstrate strong potential in low-level tasks such as super-resolution and compression. Therefore, we aim to investigate whether LLMs can be effectively used for image compression with simple visual prompts. We believe it may be more beneficial to conduct experiments using LLMs as a starting point rather than VLMs, as this approach allows us to avoid the influences of varying visual architectures present in VLMs.
> > >
> > > We hope that our response has addressed your concerns and that you will reconsider your decision.
> > >
> > > [1] Gregoire Deletang et al., Language modeling is compression. In ICLR, 2024

---

> > > > ### Author Response · Authors · 2024-12-03
> > > >
> > > > Thank you for taking the time to discuss the paper with us. We would like to know if the statement we provided on VLMs addresses your concerns. Your feedback is greatly appreciated.

---

### Official Review · Reviewer_Eb4o · 2024-11-05

**Soundness:** 3
**Presentation:** 2
**Contribution:** 2
**Rating:** 5
**Confidence:** 4

**Summary:**

The paper presents a framework for lossless image compression by leveraging Large Language Models (LLMs) combined with visual prompts. The approach generates a lossy reconstruction of an input image, extracts local and global visual features from this as prompts, and uses the LLM to predict the probability distribution of residuals for entropy coding. This visual prompt-based strategy allows LLMs to perform better on image compression tasks, even outperforming traditional codecs and several state-of-the-art learning-based methods. The framework shows notable adaptability across different image types, including medical and screen content images, demonstrating its generalization capability.

**Strengths:**

1. The paper explores a unique application of LLMs in the field of image compression, addressing the challenge of adapting LLMs from their textual foundations to image processing.
2. Incorporating lossy reconstructions as visual prompts effectively bridges the gap between textual and visual data, enhancing the LLM’s ability to predict distributions in image compression.
3. The method is validated on multiple datasets (DIV2K, CLIC, Kodak), achieving state-of-the-art compression performance.
4. Extensive ablation studies clarify the contributions of visual prompts, embedding optimization, and finetuning (using LoRA), offering a thorough analysis of each component’s impact on compression efficacy.

**Weaknesses:**

1. The pixel-by-pixel autoregressive encoding process is time-consuming, potentially limiting practical applications where computational efficiency is critical.
2. The framework’s performance is influenced by the choice and quality of the initial lossy codec. While BPG is used by default, there may be variability in performance if alternative lossy codecs are applied.
3. The proposed method’s reliance on large models and multiple GPUs could hinder scalability, especially for use cases in resource-constrained environments.
4. The code of "Large Language Model is Compression" is released. Why the auther use the reproduced preformance as the baseline?

**Questions:**

See weakness.

---

> ### Author Response · Authors · 2024-11-22
> **Response to reviewer Eb4o**
>
> Thank you for your insightful questions. We have prepared detailed, point-by-point responses to each query. We hope this addresses your concerns effectively.
>
> **Q1: About time-consuming autoregressive coding process && model complexity**
>
> **A1:** Thanks for your question. Although the autoregressive method is time-consuming, we implement a parallelization strategy to accelerate the process. Specifically, we partition the image into 16x16 patches, allowing for parallel inference across these patches and significantly reducing the inference time.
>
> We acknowledge that utilizing LLMs as entropy coding modules introduces greater complexity. However, the primary goal of this study is to demonstrate the feasibility of LLMs as effective entropy encoders and to emphasize the critical role of visual prompts in improving overall encoding efficiency. We hope this research lays a foundation for future advancements in coding systems aimed at further enhancing compression performance.
>
> We use 8 NVIDIA A100 GPUs to measure the time taken to encode one image (Kodak dataset, 768x512 resolution) and provide the results in the table below. Our experiments with LLMs of varying sizes confirm that the proposed method achieves good results even with smaller LLMs, which demonstrate that the inference time can be further shortened by a factor of 3.5. In the future, techniques such as lightweight model architectures, pruning, and quantization can be explored to reduce computational complexity without significant performance degradation.
>
>
> | Kodak           | Params | bpsp | Enc/Dec Time(second) |
> | --------------- | ------ | ---- | -------------------- |
> | Deletang et al. | 8B     | 4.84 | 10.44/288.0          |
> | Ours(1B)        | 1B+4M  | 3.24 | 3.84/141.6           |
> | Ours(3B)        | 3B+4M  | 3.21 | 10.08/338.4          |
> | Ours(8B)        | 8B+4M  | 3.19 | 21.12/495.6          |
>
>
> **Q2: About other lossy codec**
>
> **A2:** Thanks for your question. In our design, BPG serves as a replaceable lossy codec. For comparison, we replaced it with JPEG, demonstrating that JPEG can also function as a suitable lossy codec when an appropriate QP/quality is selected. Additional experiments and analyses will be included in the appendix for further detail.
>
> | Lossy Codec      | Lossy | Residual | Total |
> | ---------------- | ----- | -------- | ----- |
> | BPG(QP=34)       | 0.13  | 3.13     | 3.26  |
> | JPEG(quality=50) | 0.29  | 2.99     | 3.28  |
>
>
> **Q3: About the reproduction of [1]**
>
> **A3:** Thanks for your reminder, and we have actually noticed this. As noted in Section 4.1, "Since the LLM used in their approach is not open-source, we substitute it with LLaMA3-8B as the default model while following their other settings". [1] only released code for small transformer-based approach, which is not the main focus of this paper. Our work specifically explores the application of LLMs in image compression.
>
> [1] Gregoire Deletang et al., Language modeling is compression. In ICLR, 2024

---

> > ### Author Response · Authors · 2024-12-03
> >
> > Thank you for taking the time to review our paper. We sincerely hope that we have addressed your questions. Please let us know if you have any further questions or concerns.

---

### Official Review · Reviewer_vob6 · 2024-11-05

**Soundness:** 3
**Presentation:** 2
**Contribution:** 2
**Rating:** 6
**Confidence:** 3

**Summary:**

In this submission, the authors present a method for lossless image compression with LLMs. The main difference between the proposed one and existing LLM-based method (Deletang et al. 2023) lies in the input to LLM. Specifically, the authors show that supplying visual prompts, i.e., a lossy reconstruction, to the LLM for fine-tuning its compression capacity on the probabilistic modeling of residuals is helpful for the performance, leading to an advantage compared to modern and NN-based lossless image compression frameworks.

**Strengths:**

1. The design of supplying visual prompts and modeling only residuals is interesting and reasonable. Modeling residuals is usually easier.
2. The performance gain over the baseline is obvious. And the effectiveness of main designs, e.g., global and local prompts, embedding optimization, is proven to be positive through ablation results.

**Weaknesses:**

1. Although the experiments results on DIV2K, CLIC, Kodak are promising, the number of training images (DIV2K train split) seems limited to me. Is the method still effective if we try to extend it to larger datasets, e.g., ImageNet?

2. Since the results of the main competitor (Deletang et al. 2023) are reproduced on Kodak benchmark, why are the results are missing on other evaluation datasets?

3. How does the model size influence the compression rate? According to Deletang et al. 2023, the model size influences the compression rates in a non-monotonous way.

**Questions:**

1. How does the method perform on larger datasets?

2. Why the results on other evaluation benchmarks are missing for Deletang et al. 2023.

3. A chart like Fig. 2 in Deletang et al. 2023 is appreciated to show the trend of compression rates versus model size.

---

> ### Author Response · Authors · 2024-11-22
> **Response to reviewer vob6**
>
> Thank you for your insightful questions. We have prepared detailed, point-by-point responses to each query. We hope this addresses your concerns effectively.
>
> **Q1: About larger datasets**
>
> **A1:** Thanks for your question. The datasets we use for training were large enough. We use ImageNet2012 for the first phase of training, and DIV2K for the second stage. During training, we divide each image into 16x16 patches, generating approximately 8.64M training patches from DIV2K alone, which provide ample data for finetuning.
>
> Our test set is comprehensive, consisting of 226 images from Kodak, DIV2K, CLIC.mobile, and CLIC.pro. To further evaluate the robust of our model, we also tested it on the ImageNet validation set. Due to time constraints, we limited the evaluation to the first 200 images and achieved a result of 2.67 bpsp, surpassing the SOTA method like DLPR. These findings demonstrate the robustness and effectiveness of our approach across diverse datasets.
>
> | Codec | JPEG-XL | L3C  | DLPR | Ours     |
> | ----- | ------- | ---- | ---- | -------- |
> | bpsp  | 2.91    | 3.30 | 2.77 | **2.67** |
>
> **Q2: About more evaluation of [1]**
>
> **A2:** Thank you for your suggestion. We conduct additional evaluations of [1] on other datasets, and the results are summarized in the table below. The findings indicate that our proposed method consistently outperforms the approach in [1] by a significant margin, even without LoRA finetuning. These results underscore the robustness of our method and will be included in the final version of the paper.
>
> | bpsp           | DIV2K | CLIC.pro | CLIC.mobile | Kodak |
> | -------------- | ----- | -------- | ----------- | ----- |
> | [1]            | 4.25  | 3.99     | 4.12        | 4.84  |
> | Ours(w/o LoRA) | 2.81  | 2.71     | 2.50        | 3.19  |
> | Ours(w/ LoRA)  | 2.29  | 2.25     | 2.07        | 2.83  |
>
> **Q3: About compression ratio relative to model size**
>
> **A3:** Thank you for your question. We observe that the raw compression rate increases as the size of the LLM model increases, which is consistent with the findings in [1]. Specifically, Fig. 2 in [1] illustrates that the adjusted compression ratio (which includes model size) is non-monotonic; for the LLM, this adjusted compression ratio decreases with increasing model size, often exceeding 100%, since the model size constitutes a significant portion of the overall size. In contrast, the raw compression ratio—calculated without accounting for the model size—increases with model size. The following table lists the raw compression ratios for LLMs of different sizes.
>
> | kodak                 | Ours(1B) | Ours(3B) | Ours(8B) |
> | --------------------- | -------- | -------- | -------- |
> | bpsp                  | 3.24     | 3.21     | 3.19     |
> | Raw compression ratio | 40.5%    | 40.1%    | 39.8%    |
>
> [1] Gregoire Deletang et al., Language modeling is compression. In ICLR, 2024

---

> > ### Comment · Reviewer_vob6 · 2024-11-26
> >
> > Thanks for your responses. Most of my concerns have been addressed.

---

> > > ### Author Response · Authors · 2024-11-26
> > >
> > > Thanks for your response! We are glad to address your questions.

---

### Official Review · Reviewer_UiyY · 2024-11-05

**Soundness:** 2
**Presentation:** 2
**Contribution:** 2
**Rating:** 6
**Confidence:** 2

**Summary:**

The paper proposes to update the framework for LLM conditioned image compressions with an image or vision embedding that is fed also to the LLM, resembling in the "An Image is Worth 16x16 Words: Transformers for Image Recognition at Scale", https://arxiv.org/abs/2010.11929. The paper argues gains at the range of +34% when optimized embeddings and both local/global prompts are employed.
The model optimizes the joint entropy to measure the discrepancy between estimated and original (reference) signal. The model is trained in two stages and LLaMA3-8B is used for the language model. The first stage of training leverages ImageNet2012 with freezed LLMs and for the second stage the LLm is tuned using Lora. The model is evaluated on DIV2k, CLIC mobile validation, CLIC pro and kodak images datasets. The proposed method shows superior performance to the baseline methods.

**Strengths:**

The paper is well organized and well-written.
The experiments considers various perspectives including an ablation study.
The formulation and idea is relatively simple and straight forward.
The architecture is very similar to transformer-based encoders for image and video coding

**Weaknesses:**

Looks a bit incremental and confusing with transformer based architectures for visual data.

**Questions:**

- How does the method compares to JPEG-AI?
- The architecture resembles a transformer based methods for visual data coding, could you elaborate the differences?

---

> ### Author Response · Authors · 2024-11-22
> **Response to reviewer UiyY**
>
> Thank you for your insightful questions. We have prepared detailed, point-by-point responses to each query. We hope this addresses your concerns effectively.
>
> **Q1: About difference between Transformer-based model**
>
> **A1:** Thanks for your question. LLMs differ from ordinary ViT [1] in two significant aspects. First, ViT employs an **encoder-only** architecture that performs forward only once. In contrast, the LLM we use is a **decoder-only** architecture designed for **causal inference**, generating each token sequentially based on preceding tokens. As illustrated in Figure 4 of the Appendix, the increase in preceding tokens provides more context, allowing the model to predict the data distribution with greater accuracy.
>
> Second, LLMs are **pre-trained** models on extensive datasets, containing over **1 billion** parameters. Our objective is to investigate how LLMs pre-trained for language tasks can be effectively applied to image tasks. Although LLMs are not initially exposed to visual data, we demonstrate that their viusal potential can be unlocked through modules like visual prompts and LoRA. The causal structure of LLMs aligns with autoregressive prediction, making them highly adaptable to image-compression-related challenges.
>
> Our work introduces visual prompts into the LLM architecture via lossy compression, an aspect overlooked by previous methods. By incorporating visual prompts, our approach achieves significant performance improvements with high efficiency. We believe these contributions hold meaningful value for advancing the field of image compression.
>
> **Q2: About comparison with JPEG-AI**
>
> **A2:** Thanks for your question. To the best of our knowledge, JPEG-AI is a lossy compression method, whereas our proposed pipline is specifically designed for lossless compression. As such, a direct compression between the two is not applicable.

---

> > ### Comment · Reviewer_UiyY · 2024-11-27
> >
> > Thank you! I keep my original rating. Overall it is a nice paper. Given the overall discussions, it may benefit from reflecting some of those discussions in the paper.

---

> > > ### Author Response · Authors · 2024-11-28
> > >
> > > Thank you for your response! We have made changes to the paper, addressing both experiments and the suggestions discussed. The updates are as follows:
> > >
> > > * An introduction on arithmetic coding has been added to Section 2.2 (lines 116-121).
> > > * Experimental results for [1] and our method (without LoRA) have been included (lines 342-343 and 952-965).
> > > * We have moderated our claims regarding the role of optimized embeddings (lines 395-397).
> > > * Ablation experiments on LLM size and patch size have been added to Section 4.3 (lines 398-412).
> > > * Discussions related to model complexity have been added in Section 4.5 and Appendix E (lines 502-518 and 888-917).
> > > * Ablation experiments on the lossy codec (lines 918-935) and GMM (lines 937-950) have also been added to the Appendix.
> > >
> > > [1] Gregoire Deletang et al., Language modeling is compression. In ICLR, 2024

---

### Official Review · Reviewer_SGza · 2024-11-05

**Soundness:** 3
**Presentation:** 3
**Contribution:** 2
**Rating:** 5
**Confidence:** 5

**Summary:**

This paper presents a lossless image compression method that utilizes an LLM for entropy coding. Specifically, it employs a two-layer design where the first layer encodes the input image using lossy compression. The residual between the original image and its lossy reconstruction is then losslessly coded in the second layer using the LLM as an entropy model, with the lossy reconstructed image generating visual prompts for the LLM. To enable the LLM to better handle the distribution prediction task for entropy coding, LoRA is adopted to adapt the pre-trained LLM model.

**Strengths:**

Using LLMs for entropy coding in image compression is a relatively new topic. Experimental results show that the proposed method achieves better coding performance compared to the prior work that also adopts LLMs for entropy coding.

**Weaknesses:**

1. According to Table 2, it appears that after applying the global prompt, the additional improvement from using the local prompt is minimal, providing only a 0.8% gain. Furthermore, although the paper emphasizes the importance of optimized embedding, Table 2 shows it provides only a modest 0.4% additional gain.

2. The comparison of complexity with baseline methods (e.g., model size, encoding/decoding MACs, or runtime) is not provided.

3. The practicality of using an LLM as an entropy model is questionable, as the complexity is significantly higher than that of traditional codecs in all aspects. Furthermore, according to Tables 1 and 3, without adopting LoRA, the coding performance of the proposed method is only on par with JPEG2000, achieving better performance only with LoRA. However, LoRA fine-tuning is not novel and it also makes the LLM task-specific. As shown in Section 4.4, unlike the baseline method by Delétang et al., which does not require retraining, this approach lacks generality and necessitates retraining of the introduced components for different data domains, resulting in increased training time and storage demands.

**Questions:**

1. In Table 1, the authors mention that they reimplement the method from Delétang et al. However, why is only the Kodak dataset's result provided in Table 1? What about the results for the other three datasets?

2. It is unclear why ImageNet2012 is used for training stage 1 while DIV2K is used for training stage 2, rather than employing the same dataset throughout.

3. The motivation of using GMM is unclear. Typical learned image codecs usually utilize Gaussian distribution.

4. The model appears to be trained using 16x16 patches. Would the performance improve if the training patch size were increased?

5. It would be interesting to report results with lower or larger QP values. For instance, if the lossy branch utilizes an extremely low number of bits, resulting in poor quality of the lossy image and significant loss of original image information, can the total still remain at 3.2 bpsp? How much information can the visual embedding provide in that scenario?

---

> ### Author Response · Authors · 2024-11-22
> **Response to reviewer SGza[1/2]**
>
> Thank you for your insightful questions. We have prepared detailed, point-by-point responses to each query. We hope this addresses your concerns effectively.
>
> **Q1: About effect of optimized embedding**
>
> **A1:** Thanks for your question. The optimized embedding has a substantial impact, as shown in Table 2, using local prompts alone increases performance by 15.5%, while combining local prompts and embedding optimizing yields a 33.3% improvement----a significant enhancement. Although adding them with global prompts results in only a 0.8% further improvement, this gain is still non-trivial given the already strong baseline.
>
> | Local    | Global   | Optimized Embeddings | bpsp | Gain   |
> | -------- | -------- | -------------------- | ---- | ------ |
> | &#10004; |          |                      | 4.09 | -15.5% |
> | &#10004; |          | &#10004;             | 3.23 | -33.3% |
> | &#10004; | &#10004; | &#10004;             | 3.19 | -34.1% |
>
>
> **Q2: About complexity && practicality of using LLM as an entropy model**
>
> **A2:** Thanks for your question. Our codec process operates in parallel across patches. We use 8 NVIDIA A100 GPUs to measure the time taken to encode one image (Kodak dataset, 768x512 resolution) and provide the results in the table below. We acknowledge that utilizing LLMs as entropy coding modules introduces greater complexity compared to traditional methods.  However, the primary goal of this study is to demonstrate the feasibility of LLMs as effective entropy encoders and to emphasize the critical role of visual prompts in improving overall encoding efficiency. We hope this research lays a foundation for future advancements in coding systems aimed at further enhancing compression performance.
>
> Our experiments with LLMs of varying sizes confirm that the proposed method achieves good results even with smaller LLMs, significantly reducing computational time. In the future, techniques such as lightweight model architectures, pruning, and quantization can be explored to reduce computational complexity and consuming time without significant performance degradation.
>
> | Kodak           | Params | bpsp | Enc/Dec Time(second) |
> | --------------- | ------ | ---- | ------------ |
> | Deletang et al. | 8B     | 4.84 | 10.44/288.0  |
> | Ours(1B)        | 1B+4M  | 3.24 | 3.84/141.6   |
> | Ours(3B)        | 3B+4M  | 3.21 | 10.08/338.4  |
> | Ours(8B)        | 8B+4M  | 3.19 | 21.12/495.6  |
>
>
> **Q3: About effect of LoRA**
>
> **A3:** Thanks for your question. LoRA is indeed a widely used module; however, our primary objective is to explore the potential of language-pretrained LLMs for image compression tasks, with LoRA not being the main contribution of this paper.
>
> While [1] does not require training, our experiments show that by integrating residuals, visual prompts, and embedding retraining, we achieve a 34.1% improvement over [1], with an additional 11.0% boost from LoRA. These results highlight the effectiveness of our approach.
>
> **Q4: About more evaluation of [1]**
>
> **A4:** Thanks for your suggestion. We further evaluate [1] on additional datasets, and the results are presented in the table below. The findings indicate that the method proposed in this paper consistently outperforms the approach in [1] by a significant margin. These results will be included in the final version of the paper.
>
> | bpsp           | DIV2K | CLIC.pro | CLIC.mobile | Kodak |
> | -------------- | ----- | -------- | ----------- | ----- |
> | [1]            | 4.25  | 3.99     | 4.12        | 4.84  |
> | Ours(w/o LoRA) | 2.81  | 2.71     | 2.50        | 3.19  |
> | Ours(w/ LoRA)  | 2.29  | 2.25     | 2.07        | 2.83  |
>
> [1] Gregoire Deletang et al., Language modeling is compression. In ICLR, 2024
>
> **Q5: About training strategy**
>
> **A5:** Thank you for your question. In the first training stage, we train the Global Embedding Module, which requires full images and a large dataset for effective training. To fulfill this reqirement, we choose ImageNet2012 due to its extensive quantity of images, despite their relatively lower resolution. In the second stage, we train LoRA to further enhance the LLM's performance, focusing mainly on patches rather than whole images. Replacing ImageNet2012 with the higher-resolution DIV2K dataset during this stage improve performance from 2.92 bpsp to 2.83 bpsp. Although DIV2K consists only 800 images, its high resolution provides 8.64 million patches, offering ample data for LoRA training and ensuring effective optimization.

---

> > ### Author Response · Authors · 2024-11-22
> > **Response to reviewer SGza[2/2]**
> >
> > **Q6: About GMM**
> >
> > **A6:** Thank you for the question. GMM is commonly used in image compression, such as [2] [3]. Compared to the Gaussian single model (GSM), GMM introduces minimal parameters while providing moderate improvements. Our ablation study on the number of mixtures K in GMM indicates that K=5 significantly outperforms K=1, highlighting its superior ability to capture complex distributions effect.
> >
> > | BPSP  | K=1  | K=5  |
> > | ----- | ---- | ---- |
> > | Kodak | 3.29 | 3.19 |
> >
> > [2] Zhengxue Cheng et al., Learned Image Compression With Discretized Gaussian Mixture Likelihoods and Attention Modules. In CVPR, 2020
> >
> > [3] Yuanchao Bai et al., Deep lossy plus residual coding for lossless and near-lossless image compression. IEEE Transactions on Pattern Analysis and Machine Intelligence, 2024
> >
> > **Q7: About the effect of patch size**
> >
> > **A7:** Thanks for your question. We originally use a patch size of 16x16 and extend our experiments to 24x24. We find that increasing the patch size leads to slight performance improvements, as longer contexts contain more information, resulting in better compression. Regarding the 32x32 experiments, our device has limited memory resources, preventing us from conducting further evaluations at this time. The table below presents the outcomes for different patch sizes:
> >
> >
> > | BPSP  | 16x16 | 24x24 |
> > | ----- | ----- | ----- |
> > | Kodak | 3.19  | 3.16  |
> >
> >
> > **Q8: About the effect of BPG QP**
> >
> > **A8:** Thanks for your question. Our findings reveal that, within an appropriate QP range, the choice of QP has a limited effect on overall performance. However, extreme QP settings can lead to significant performance degradation.
> >
> > We expand the range of QP values selected for BPG. The experiments show that when QP values range between [22, 34], the bpsp remains relatively stable. However, setting the QP value too low (e.g., QP=14) reduces performance, as the bitrate required for lossy coding increases significantly. Conversely, setting the QP value too high (e.g., QP=42) also degrades performance due to the excessive residuals that must be compressed losslessly. We will add more detailed experiments and analyses in the final version of the paper.
> >
> > | Lossy Codec | Lossy | Residual | Total |
> > | ----------- | ----- | -------- | ----- |
> > | BPG(QP=14)  | 0.95  | 2.43     | 3.38  |
> > | BPG(QP=22)  | 0.48  | 2.72     | 3.20  |
> > | BPG(QP=28)  | 0.27  | 2.92     | 3.19  |
> > | BPG(QP=34)  | 0.13  | 3.13     | 3.26  |
> > | BPG(QP=42)  | 0.04  | 3.38     | 3.42  |

---

> > > ### Comment · Reviewer_SGza · 2024-11-25
> > >
> > > Thanks for the authors' reply. Below are the following questions:
> > > 1. **[Effectiveness of local and optimized embeddings]** Based on the authors’ response and Table 2, the reported gains of local prompts, global prompts, and optimized embeddings depend on the order in which these components are introduced. Although the authors emphasize during rebuttal that local prompts and optimized embeddings are important, Table 2 shows that using only the global prompts (without local prompts or optimized embeddings) already achieved a significant -32.9% BD-rate reduction, representing the most substantial improvement. Adding local prompts and optimized embeddings provides only a minor additional gain of 1.2% (-32.9% to -34.1%), with 0.8% from the local prompt (-32.9% to -33.7%) and 0.4% from the optimized embedding (-33.7% to -34.1%). These additional gains are limited and not as impactful as claimed. Given the modest improvements compared to the increased training and inference costs, the global prompt alone may suffice.
> > >
> > >
> > > 2. **[Complexity comparison]** A complexity comparison with non-LLM-based learning-based lossless image compression methods is not provided.
> > >
> > > 3. **[Practicality of Using LLM as an Entropy Model]** The practicality of using LLM as an entropy model remains questionable. As shown in Table 1, DLPR achieves comparable performance to the proposed method but likely with much lower complexity. While performance gains can come at the cost of increased complexity, a slight increase in DLPR’s complexity is possible to surpass the proposed method's performance. In contrast, the smallest model of the proposed method takes 3.84 seconds to encode one image using 8 NVIDIA A100 GPUs, which is extremely impractical. Although the authors suggest that future strategies like pruning and quantization can reduce complexity, these often compromise performance. Given the proposed method’s high baseline complexity and its relatively limited performance—only comparable to JPEG2000 without LoRA and to JPEG-XL with LoRA—it is uncertain how much performance can be maintained if complexity is reduced to a more practical level. Since the provided inference complexity requires 8 NVIDIA A100 GPUs, another follow-up question is that the training complexity of the proposed method is not provided in the paper.
> > >
> > > 4. **[About the effect of BPG QP]** It is suggested that more explanation be provided on why both extremely low and high QP settings lead to significant performance degradation in the proposed method.

---

> > > > ### Author Response · Authors · 2024-11-26
> > > >
> > > > Thanks for your reply, we will reply to your questions point by point.
> > > >
> > > > **Q1: About effectiveness of local and optimized embeddings.**
> > > >
> > > > **A1:** Our local modules essentially function as lookup tables, introducing minimal additional training and inference costs. The optimized embeddings also entail very little training overhead. Both techniques contribute to performance improvements. And the mainly complexity of our approach stems from the LLM itself. If we pursue a simpler architecture, utilizing only global modules would be sufficient, but local and optimized embeddings also make some contribution. We will revise the statements in the paper to weaken our claims regarding local modules and optimized embeddings.
> > > >
> > > > **Q2: About complexity comparison.**
> > > >
> > > > **A2:** We further provide the runtime of other non-LLM-based learning-based methods in the table below.
> > > >
> > > > | Kodak           | Params | Enc/Dec Time  |
> > > > | --------------- | ------ | ------------- |
> > > > | L3C             | 5M     | 8.17s/7.89s   |
> > > > | DLPR            | 37M    | 1.26s/1.80s   |
> > > > | Deletang et al. | 8B     | 10.44s/288.0s |
> > > > | Ours(1B)        | 1B+4M  | 3.84s/141.6s  |
> > > > | Ours(3B)        | 3B+4M  | 10.08s/338.4s |
> > > > | Ours(8B)        | 8B+4M  | 21.12s/495.6s |
> > > >
> > > > **Q3: About practicality of Using LLM as an Entropy Model.**
> > > >
> > > > **A3:** We acknowledge that our solution may not currently prioritize practicality; instead, our focus lies in its academic value. The widespread success of LLMs across various fields promopts us to explore their potential contributions to image compression tasks. We demonstrate that LLMs can easily achieve state-of-the-art performance, indicating substantial unapped potential in LLM-based architectures. We believe this direction holds significant academic value for advancing image compression. Our work aims to inspire more researchers to engage in this field, driving improvements inperformance of LLM-based codec, reducing complexity, and ultimately making LLMs practical image codecs.
> > > >
> > > > We utilize 8 A100-40G GPUs primarily to enable parallel inference, thereby accelerating the encoding and decoding processes. However, our method is also compatible with GPUs with as little as 14G of memory. For training, we employ 4 A100-40G GPUs over three days, although the process can also be completed on a single 24G GPU, albeit with a longer training time.
> > > >
> > > >
> > > > **Q4: About the effect of BPG QP.**
> > > >
> > > > **A4**: When the QP is lower, the quantization step (Qstep) becomes smaller, leading to an improvement in its quality and an increase in the bpsp of the lossy image. This results in a simpler residual, which requires a lower bpsp for compression. However, the increase in bpsp for the lossy image outweighs the decrease in bpsp for the residual, leading to an overall increase in the total bpsp.
> > > >
> > > > When the QP is higher, the quantization step (Qstep) becomes bigger, leading to an decrease in its quality and an reduction in the bpsp of the lossy image. This results in a complex residual, which requires a bigger bpsp for compression. However, the reduction in bpsp for the lossy image outweighs the increase in bpsp for the residual, leading to an overall increase in the total bpsp.

---

> > > > > ### Author Response · Authors · 2024-12-03
> > > > >
> > > > > Thank you for your further suggestions and questions. We have revised the claim about embeddings in our paper and included the relevant data to address the rest of your concerns. We sincerely appreciate your insights and would be grateful if you could let us know if our responses have addressed your concerns.

---

### Official Review · Reviewer_H5zc · 2024-11-08

**Soundness:** 3
**Presentation:** 3
**Contribution:** 3
**Rating:** 8
**Confidence:** 5

**Summary:**

The authors present a system that utilizes the pattern recognition ability of LLM's to extract accurate probability models to perform lossless entropy coding of real world images. To improve the performance, they introduce a host of methods and tricks, following it up with appropriate ablations. This results in a system that beats standard methods like PNG, JPEG-XL in image compression.

**Strengths:**

- Instead of directly encoding the image pixels like previous works in this domain, they use off the shelf lossy compression models and encode their residuals. To achieve this, they obtain embeddings for the entire image (global), patches (local) and use it as "prompts" to the language model, along with a GMM (Gaussian mixture model) to accurately model the underlying distribution.
- The proposed system might appear to be a hodge·podge of different tricks, but each one contributes meaningfully towards arriving at the solution. This is highlighted in their extensive ablation study.
- The flow of the paper is good. I like the simplicity of the approach - with no unnecessary additions or jargon which has become the bane of our community.

**Weaknesses:**

- The most obvious weakness of such a system would be the required compute. Hence, I think a mention about the MACs for encoding and decoding would be good. I wonder if the authors tried any Quantized LLM's for this task - something that is known to be small and occupy much smaller footprints. Eg: [Phi 3 mini](https://huggingface.co/QuantFactory/Phi-3-mini-128k-instruct-GGUF) or even [BitNet](https://github.com/microsoft/BitNet).
- For a system that relies on an LLM for most of the heavy lifting, I am a tad bit disappointed to not see a study that ablates on them. I am particularly intrigued if there is any correlation between LLM size and their encoding ability.
- On similar lines, I would love to see ablations on image encoding as well. It would be helpful to try out different image encoders, starting with something basic like JPEG.

**Questions:**

Is there a particular reason for choosing BPG as the image encoder, when you could have started with something cheap like JPEG (maybe at quality 50) ?

---

> ### Author Response · Authors · 2024-11-22
> **Response to reviewer H5zc**
>
> Thank you for your insightful questions. We have prepared detailed, point-by-point responses to each query. We hope this addresses your concerns effectively.
>
> **Q1: About kMACs && complexity && Quantized LLM**
>
> **A1:** Thanks for your question. The per-pixel kMACs of our method are presented in the table below. Although our method requires a large number of kMACs, using a smaller LLM (1B/3B) significantly reduces kMACs with minimal performance degradation. We also observe that as the size of the LLM increases, the compression ratio improves. Notably, our method based on the 1B LLM outperforms existing solutions, demonstrating its effectiveness despite the reduced model size.
>
> | Codec           | Enc/Dec kMACs/pixel | bpsp | Compression Ratio |
> | --------------- | ------------------- | ---- | ----------------- |
> | Deletang et al. | $2.1\times10^7$     | 4.84 | 60.5%             |
> | Ours(1B)        | $5.9\times10^6$     | 3.24 | 40.5%             |
> | Ours(3B)        | $1.7\times10^7$     | 3.21 | 40.1%             |
> | Ours(8B)        | $4.2\times10^7$     | 3.19 | 39.8%             |
>
>
> We further investigate quantizing LLaMA3-8B using four methods: BitsAndBytes [1], GPTQ [2], AWQ [3], and BitNet [4]. The test results on the Kodak dataset are presented. Our findings indicate that 8-bit quantization has a negligible impact on performance, while 4-bit quantization (via AWQ) still achieves relatively good results. However, significant performance degradation is observed with 1-bit quantization due to its substantial impact on the model's representation capability. Additionally, directly replacing LLaMA3-8B with BitNet (LLaMA3-8B-Instruct), which is not aligned with our additional trained modules, further exacerbates the performance loss. In future work, we aim to explore advanced quantization techniques to mitigate such performance losses.
>
>
> | Kodak                | bpsp |
> | -------------------- | ---- |
> | Deletang et al.      | 4.84 |
> | Ours                 | 3.19 |
> | Ours(bnb 8bit)       | 3.23 |
> | Ours(bnb 4bit)       | 4.00 |
> | Ours(GPTQ 4bit)      | 3.65 |
> | Ours(AWQ 4bit)       | 3.43 |
> | Ours(BitNet 1.58bit) | 5.71 |
>
>
> [1] Dettmers, Tim, et al. "Gpt3. int8 (): 8-bit matrix multiplication for transformers at scale." Advances in Neural Information Processing Systems 35 (2022): 30318-30332.
>
> [2] Frantar, Elias, et al. "Gptq: Accurate post-training quantization for generative pre-trained transformers." arXiv preprint arXiv:2210.17323 (2022).
>
> [3] Lin, Ji, et al. "AWQ: Activation-aware Weight Quantization for On-Device LLM Compression and Acceleration." Proceedings of Machine Learning and Systems 6 (2024): 87-100.
>
> [4] Ma, Shuming, et al. "The era of 1-bit llms: All large language models are in 1.58 bits." arXiv preprint arXiv:2402.17764 (2024).
>
>
> **Q2: About ablation on lossy image codec**
>
> **A2:** Thanks for your question. We expand the range of QP values selected for BPG and introduce JPEG as an additional lossy codec. The lossy codec is flexible and can be replaced with different codecs with appropriate QP settings, underscoring the adaptability and robustness of our overall framework.
>
> The experiments of BPG show that when QP values range between [22, 34], the bpsp remains relatively stable. However, setting the QP value too low (e.g., QP=14) reduces performance, as the bitrate required for lossy coding increases significantly. Conversely, setting the QP value too high (e.g., QP=42) also degrades performance due to the excessive residuals that must be compressed losslessly. Similar trends are observed for JPEG; when an appropriate quality is selected, our framework consistently maintains high performance. We will add more detailed experiments and analyses in the final version of the paper.
>
> | Lossy Codec      | Lossy | Residual | Total |
> | ---------------- | ----- | -------- | ----- |
> | BPG(QP=14)       | 0.95  | 2.43     | 3.38  |
> | BPG(QP=22)       | 0.48  | 2.72     | 3.20  |
> | BPG(QP=28)       | 0.27  | 2.92     | 3.19  |
> | BPG(QP=34)       | 0.13  | 3.13     | 3.26  |
> | BPG(QP=42)       | 0.04  | 3.38     | 3.42  |
> | JPEG(quality=30) | 0.20  | 3.30     | 3.50  |
> | JPEG(quality=50) | 0.29  | 2.99     | 3.28  |
> | JPEG(quality=70) | 0.40  | 2.96     | 3.36  |

---

### Official Review · Reviewer_GMiU · 2024-11-08

**Soundness:** 3
**Presentation:** 4
**Contribution:** 4
**Rating:** 8
**Confidence:** 4

**Summary:**

The authors address the problem of lossless image compression by proposing a framework that leverages a Large Language Model as the entropy model to estimate the distribution of residuals between the original image and its lossy-compressed counterpart. Extensive ablation studies demonstrate the framework’s advantages over previous methods.

**Strengths:**

Building on previous work that uses Large Language Models (LLMs) as entropy models for lossless compression, the authors argue that the limited performance gains over traditional (non-learning-based) methods stem from differences between the textual features captured by pre-trained LLMs and the intrinsic characteristics of image pixels. To address this, they propose inputting visual embeddings into the LLM to enhance performance. The concept is clearly presented, the argument is compelling, and the experiments are thoroughly conducted.

**Weaknesses:**

The authors acknowledge the time-consuming nature of their proposed approach. However, it is important to note that this issue arises not only from the autoregressive structure of the method but also from the additional computational load introduced by the visual embeddings compared to previous approaches. A comparison of this aspect would be insightful. The authors also address the role of lossy compression within the framework. They conducted an ablation study on the quantization parameter of BPG for lossy compression, showing that this parameter does not affect the overall performance. However, further explanations of this effect would be valuable.

**Questions:**

N/A

---

> ### Author Response · Authors · 2024-11-22
> **Response to reviewer GMiU**
>
> Thank you for your insightful questions. We have prepared detailed, point-by-point responses to each query. We hope this addresses your concerns effectively.
>
> **Q1: About the additional computational load from the visual embeddings.**
>
> **A1:** Thanks for your question. We separately calculate the computational overhead introduced by the visual embedding module and the LLM. The results show that the additional parameters from the visual prompts module account for only a small fraction of the total and have a negligible impact on the overall kMACs.
>
> | Module      | Visual Embedding | LLM             |
> | ----------- | ---------------- | --------------- |
> | kMACs/pixel | $1.1\times10^3$  | $4.2\times10^7$ |
> | Params      | 4M               | 8B              |
>
> **Q2: About the effect of BPG QP**
>
> **A2:** Thanks for your question. We expand the range of QP values selected for BPG and introduce JPEG as an additional lossy codec. Our findings reveal that, within an appropriate QP range, the choice of QP has a limited effect on overall performance. However, extreme QP settings can lead to significant performance degradation.
>
> The experiments of BPG show that when QP values range between [22, 34], the bpsp remains relatively stable. However, setting the QP value too low (e.g., QP=14) reduces performance, as the bitrate required for lossy coding increases significantly. Conversely, setting the QP value too high (e.g., QP=42) also degrades performance due to the excessive residuals that must be compressed losslessly. Similar trends are observed with JPEG; when an appropriate quality is selected, our framework consistently maintains high performance. We will add more detailed experiments and analyses in the final version of the paper.
>
> | Lossy Codec      | Lossy | Residual | Total |
> | ---------------- | ----- | -------- | ----- |
> | BPG(QP=14)       | 0.95  | 2.43     | 3.38  |
> | BPG(QP=22)       | 0.48  | 2.72     | 3.20  |
> | BPG(QP=28)       | 0.27  | 2.92     | 3.19  |
> | BPG(QP=34)       | 0.13  | 3.13     | 3.26  |
> | BPG(QP=42)       | 0.04  | 3.38     | 3.42  |
> | JPEG(quality=30) | 0.20  | 3.30     | 3.50  |
> | JPEG(quality=50) | 0.29  | 2.99     | 3.28  |
> | JPEG(quality=70) | 0.40  | 2.96     | 3.36  |

---

> > ### Comment · Reviewer_GMiU · 2024-11-27
> >
> > Thank you for the extra evaluations. My comments are very well addressed.

---

> > > ### Author Response · Authors · 2024-11-27
> > >
> > > Thanks for your feedback! We are pleased to address your questions.

---

### Comment · Area_Chair_nrgo · 2024-11-25

Hi Reviewers,

We are approaching the deadline for author-reviewer discussion phase. Authors has already provided their rebuttal. In case you haven't checked them, please look at them ASAP. Thanks a million for your help!

---

### Author Response · Authors · 2024-11-25

Dear Reviewers and Area Chairs,

We apperaicate the reviewers(**R1** GMiU, **R2** H5zc, **R3** SGza, **R4** UiyY, **R5** vob6, **R6** Eb4o, **R7** RowH, **R8** htWV, and **R9** GpA4) for their insightful feedback. The reviewers agree that:

**Noval approach**:

* **R3**: "Using LLMs for entropy coding in image compression is a relatively **new topic**."
* **R6**: "The paper explores a **unique** application of LLMs in the field of image compression, addressing the challenge of adapting LLMs from their textual foundations to image processing."
* **R8**: "The idea of using LLM for entropy coding is **novel** and presents a new research direction."

**Effectiveness**:
* **R3**: "Experimental results show that the proposed method achieves **better coding performanc**e compared to the prior work that also adopts LLMs for entropy coding."
* **R4**: "The proposed method shows **superior performance** to the baseline methods."
* **R5**: "The performance **gain** over the baseline is **obvious**. And the **effectiveness** of main designs, e.g., global and local prompts, embedding optimization, is proven to be positive through ablation results."
* **R6**: "Incorporating lossy reconstructions as visual prompts **effectively** bridges the gap between textual and visual data, enhancing the LLM’s ability to predict distributions in image compression."
* **R7**: "These features, alongside the residuals between the original and lossy images, guide the LLM in predicting probability distributions for the residuals, **effectively** functioning as an entropy model."
* **R9**: "The method combines traditional lossy reconstruction and LLM, showing an **effective** way for lossless image compression."

**Interesting**:
* **R5**: "The design of supplying visual prompts and modeling only residuals is **interesting** and **reasonable**. Modeling residuals is usually easier."
* **R7**: "The research direction LLM-based image compression is **interesting**."

**Well-Written and Organized**:
* **R1**: "The concept is **clearly presented**, the argument is **compelling**, and the experiments are **thoroughly conducted**."
* **R2**: "The **flow** of the paper is **good**."
* **R4**: "The paper is **well organized** and **well-written**."

For the questions raised by the reviewers (e.g., computational complexity, more detailed ablations, related concepts, and so on), we have responded individually to each reviewer to address any concerns.

Best Regards,

Authors

---

### Meta-Review · Area_Chair_nrgo · 2024-12-21

**Metareview:**

This paper works on lossless image compression. Authors proposed to use LMMs with visual prompts for lossless image compression. They first generated a lossy reconstruction of the input image as visual prompts. From which they extracted local and global features to serve as visual embeddings for the LLM. The residual between the original image and the lossy reconstruction is then fed into the LLM along with the visual embeddings, enabling LLM to act as an entropy model to predict the probability distribution of the residual. Experimental results show the effectiveness of the proposed methods. Authors also extended the work to medical and screen content images.

This paper was reviewed by 9 reviewers and got mixed scores as two 8, four 5, two 6, one 3.

Strength and weaknesses given by reviewers before rebuttals are as follows (notes that different reviewers has different perspectives of the paper, so conflicts in strength and weaknesses might happen):

Strength: 1) paper is well written; 2) argument is verified by experiments; 3) using LLMs for entropy coding in image compression is new topic; 4) the design is interesting and reasonable; 5) the performance gain over the baseline is obvious; 6) the research is interesting; 7) idea is novel; 8) ablation studies are extensive and complete;

Weaknesses: 1) require a lot of compute and need comparison of complexity with baseline methods; 2) the practicality of using an LLM as an entropy model is questionable; 3) a bit incremental and confusing with transformer based architectures for visual data; 4) whether effective for larger datasets; 5) The framework’s performance is influenced by the choice and quality of the initial lossy codec; 6) The proposed method’s reliance on large models and multiple GPUs could hinder scalability; 7) its design elements appear largely incremental, limiting the overall novelty;

During author-reviewer discussion phase:

Reviewer GMiU (rating 8) suggested their concerns are very well addressed.

Reviewer H5zc (rating 8) didn't reply.

Reviewer SGza (rating 5) suggested most of the gain is from global prompt and a complexity comparison with non-LLM-based learning-based lossless image compression methods is not provided. then authors added later. but the reviewer didn't reply back. Reviewers showed concerns on the practicality of using LLM as an entropy model.

Reviewer UiyY (rating 6) kept the rating.

Reviewer vob6 (rating 6) suggested that most concerns were addressed.

Reviewer Eb4o (rating 5) didn't reply.

Reviewer RowH (rating 5) didn't reply.

Reviewer htWV (rating 3) suggested they were not fully convinced that this is the right direction to go and mainly concern was that its practicality was questionable.

Reviewer GpA4 (rating 5) mentioned their concerns are partially addressed and asked additional questions. The authors replied. but reviewers didn't reply back.

Reviewers didn't provide any comment in the reviewer-AC discussion phase.

The main concerns from reviewers are that using LLM for image compression is not practical. Given several reviewers echoed this, AC decided to reject this paper.

**Additional Comments On Reviewer Discussion:**

During author-reviewer discussion phase:

Reviewer GMiU (rating 8) suggested their concerns are very well addressed.

Reviewer H5zc (rating 8) didn't reply.

Reviewer SGza (rating 5) suggested most of the gain is from global prompt and a complexity comparison with non-LLM-based learning-based lossless image compression methods is not provided. then authors added later. but the reviewer didn't reply back. Reviewers showed concerns on the practicality of using LLM as an entropy model.

Reviewer UiyY (rating 6) kept the rating.

Reviewer vob6 (rating 6) suggested that most concerns were addressed.

Reviewer Eb4o (rating 5) didn't reply.

Reviewer RowH (rating 5) didn't reply.

Reviewer htWV (rating 3) suggested they were not fully convinced that this is the right direction to go and mainly concern was that its practicality was questionable.

Reviewer GpA4 (rating 5) mentioned their concerns are partially addressed and asked additional questions. The authors replied. but reviewers didn't reply back.

Reviewers didn't provide any comment in the reviewer-AC discussion phase.

The main concerns from reviewers are that using LLM for image compression is not practical. Given several reviewers echoed this, AC decided to reject this paper.

---

### Decision · Program_Chairs · 2025-01-22

Reject